# Navigating tenses in Bengali sentences: A stacked ensemble model for enhanced prediction

Umme Ayman[1], Md. Nahid Hasan[2*], Ms. Nusrat Khan[1], Ms. Chayti Saha[1], Md. Fayejullah[3]

**1** Department of CSE, Daffodil International University Daffodil Smart City (DSC), Birulia, Savar, Dhaka, Bangladesh, **2** Department of CSE, Dhaka International University, Dhaka, Bangladesh, **3** Department of CSE, University of Wollongong, Wollongong, New South Wales, Australia

* nahid.hasan.bondhan@gmail.com

**Data availability statement:** The dataset supporting the findings of this study is publicly available in Mendeley Data at https://doi.org/10.17632/w9mdy6tw84.1.

## Abstract

Tense classification in Bengali sentences is a fundamental yet unsolved problem of Bangla natural language processing (NLP) which is essential for tasks like machine translation, sentiment analysis, grammar correction, writing assistance and sentence generation. This study addresses this gap by proposing a robust stacked ensemble model designed for accurate automatic tense classification in Bengali sentences. To support this, we construct a novel Bengali corpus "BengaliTenseCorpus" comprising 13,500 manually collected and meticulously labelled sentences, categorized into three tense classes: Present (0), Past (1) and Future (2). The sentences gathered from diverse sources including news articles, songs, poems and novels, went through rigorous pre-processing techniques to preserve linguistic integrity and improve performance on data. The proposed architecture integrates predictions from five base models— Random Forest, Support Vector Machine, XGBoost classifier, Long Short-Term Memory (LSTM), and Gated Recurrent Unit (GRU) into the meta model— neural network to build a stacked ensemble framework. Experimental results demonstrate that this ensemble model outperforms individual models, achieving a classification accuracy of 85% on test data. This work presents the first large-scale Bengali tense classification system combining machine learning and deep learning methods in a stacked ensemble framework, establishing a strong performance benchmark for Bangla NLP with practical applications in intelligent writing tools, grammar assistance, and language learning. The findings highlight how well ensemble-based systems can capture the intricacies of Bengali verb morphology. To further boost the development of Bangla language models and applications, future extensions of this work may involve expanding the dataset, exploring transformer-based models, and incorporating tense-to-tense morphological conversion.

## Introduction

Language is a system of conventional spoken, manual (signed), or written symbols humans use to express themselves [1]. Approximately 7,111 living languages exist today. Bangla is one

**Funding:** The author(s) received no specific funding for this work.

**Competing interests:** The authors have declared that no competing interests exist.

of the most significant languages in the world. Pali and Magdhi Prakrit are the languages that gave rise to Bengali [2]. One of the most Eastern Indo-Aryan languages spoken in the Indian subcontinent is Bengali. The Bangla language ranked 7th for the most-used speaking language in the world, with 272.8 million native speakers [3]. Despite its prevalence, Bengali's complex grammatical structure presents significant challenges for computational tasks, particularly in the area of natural language processing (NLP). One critical aspect of Bengali language is its tense system, which is essential for conveying temporal context and ensuring grammatical accuracy [4]. Tenses denote the time reference of verbs [5], allowing speakers to express actions relative to the present, past, or future [6]. Accurate tense classification is crucial for a range of NLP applications, including text classification, machine translation, and sentiment analysis [7]. This research focuses specifically on automatic tense classification in Bengali text, a core NLP classification problem aimed at improving the understanding of temporal expressions in Bangla sentences. In Bengali, tense classification is particularly challenging due to the language's intricate morphology and syntax, making it difficult for humans and machines to automate tense identification and classification accurately. Bengali has unique difficulties for NLP tasks like tense categorization since it is a low-resource language with a rich morphology. The language features complex inflectional morphology, where verb forms change based on tense, aspect, person, number, and honorifics, making rule-based or simplistic statistical models ineffective. Bengali mostly uses suffix modifications to indicate tense, which can be nuanced and context-dependent, in contrast to English, where auxiliary verbs are frequently used to indicate tense. Furthermore, it is more difficult to correctly detect verb locations and tense indicators due to its variable word order and syntactic ambiguities. These problems are made worse by the dearth of annotated language materials and tools. Our suggested stacked ensemble model is especially well-suited to address these challenges by utilizing deep learning models (LSTM and GRU) to capture the temporal dependencies and sequential patterns present in Bengali verb usage, as well as conventional classifiers (like SVM and XGBoost) for pattern recognition. In a linguistically complex language like Bengali, this hybrid technique improves the model's generalization across a variety of sentence structures and verb conjugations, leading to more accurate tense categorization. Thus, automatic tense classification from Bengali text would be the most remarkable part of the Bangla natural language processing research domain in terms of text classification. Therefore, the clear objective of this study is to develop a stacked ensemble model combining deep learning (LSTM, GRU) and machine learning (SVM, XGBoost) techniques to accurately classify tenses in Bengali text, addressing the linguistic complexities and resource limitations inherent in the Bengali that can enhance downstream applications such as machine translation, educational technologies, and AI-powered communication tools for Bengali-speaking communities. Breaking new ground in Bangla NLP, our pioneering study is the first to tackle tense classification, setting a precedent in linguistic research. Numerous academics have completed various language processing tasks to classify or categorise texts into numerous aspects of natural language processing (NLP) by applying various methods. Kumar Das et al. (2021) [8] study on Bengali hate speech detection on Facebook found that machine learning models, including an attention-based decoder, achieved a 77% accuracy rate, indicating potential for further research. Rahman and Chakraborty (2021) [9] study uses a deep recurrent neural network with BiLSTM architecture to classify Bangla documents, achieving high accuracy of 98.87%, but highlighting limitations like data scarcity. Ghosh et al. (2022) [10] created a machine learning model for detecting depressive social media texts in Bangla, achieving 94.3% accuracy, highlighting the need for diverse datasets and improvements in low-resource languages. Rezaul et al. (2023) [11] conducted a multi-class sentiment analysis on Bengali social media comments

using a dataset of 42,036 comments. They used a supervised deep learning classifier, achieving an accuracy of 85.8% and an F1 score of 0.86. Rahman et al. (2024) [12] used machine learning techniques to analyze sentiment in eLearning texts in Bangla and Romanized Bangla. They achieved high accuracy with XLM-RoBERTa, but faced limitations due to cultural context. Despite these advancements in text classification in various aspects, there is a clear gap in the Bengali NLP landscape, particularly concerning the automatic classification of tenses in Bengali text. Accurate identification and classification of verb tenses are essential for various applications in Bangla language processing and computational linguistics. The complexity of Bengalis verb forms, compounded by its rich morphology, demands more robust and specialised approaches. By focusing on this significance challenges , the research aims to bridge the gap by developing an ensemble model using machine learning and deep learning-based models by utilising a BengaliTense dataset consisting of 13,500 Bengali sentences collected from diverse sources like, Bangla blogs, Facebook pages, magazines and news articles and labeling them into three different classes to employ various data preprocessing techniques over it . The dataset contains Bengali sentences from various sources, labelling them into three classes. The model uses various data preprocessing techniques and models to enhance automatic tense classification in Bengali text. This study aims to fill gaps in existing text classification processes and expand the scope of text analysis tasks in any language related to tense classification. There is little to no work for Bangla tense categorization in the Bangla NLP job. The absence of an appropriate dataset is the main cause of this circumstance. Our unique contribution is the creation of a prediction model to forecast the tense of Bangla text and a dataset to describe the Bangla tense. All relevant data are available from the Mendeley repository at: https://data.mendeley.com/datasets/w9mdy6tw84/1.

The major contributions of this study are mentioned as follows:

- Introducing a dataset "BengaliTenseCorpus: A comprehensive corpus in Bengali texts categorized in Present , Past, and Future" which consists of 13,500 Bengali sentences collected from diverse sources categorised into three classes as Present (0), Past (1), and Future (2). "https://data.mendeley.com/datasets/w9mdy6tw84/1"
- Several pre-processing techniques have been applied to ensure data quality as well as machine feasibility.
- Machine learning and deep learning models have been studied and selected based on the type of our dataset.
- An ensemble is introduced by the combination of three selected machine learning algorithms, XGB,RF,and SVM , as well as two deep learning algorithms , LSTM and GRU.
- A comparative study is accomplished by analysing the performance of the applied models and our proposed model based on several performance metrics to identify the best performing approach.

## Literature review

Kumar Das et al. (2021) [8] investigated hate speech detection in Bengali social media comments, focusing on Facebook. The dataset contains 7,425 comments categorized into seven classes, including hate speech, religious hatred, and political comments. The study used several machine learning models, with the attention-based decoder achieving the best accuracy of 77%. Limitations include the need for improved accuracy and a larger dataset. This research provides a foundation for further work on Bengali hate speech detection using advanced ML techniques.

Asghar et al. (2021) [13] introduced the Senti-eSystem, a sentiment-based system combining fuzzy logic and deep neural networks to measure customer satisfaction. Using a dataset

of 11,541 tweets from Kaggle, categorized into positive and negative sentiments, the system employs a BiLSTM model with an attention mechanism, achieving 92.86% accuracy, outperforming traditional lexicon-based methods. Limitations include the imbalanced dataset, which could affect performance, and the need for testing across various domains. The authors also note that the dataset size is relatively small for big data applications, suggesting the potential for improvement with larger, more balanced datasets.

Rahman and Chakraborty (2021) [9] present a method for classifying Bangla documents using a Deep Recurrent Neural Network with a BiLSTM architecture. They employed a dataset of 40,000 Bangla news articles, categorized into 12 classes, after data cleaning. The model achieved a high accuracy of 98.87%, along with precision, recall, and F1 scores of 0.989. Despite these results, the study highlights limitations like the scarcity of Bangla datasets and the potential drop in performance with larger datasets.

Salehin et al. (2021) [14] conducted a comparative study on various text classification approaches for Bangla news articles, using a large dataset of 75,951 articles categorized into 12 classes. The study implemented several machine learning classifiers and neural networks, with LSTM achieving the highest accuracy at 87%, followed by SVM and XGBoost at 77%. Despite these promising results, the paper notes limitations, such as the use of a single dataset type and the risk of overfitting with complex models like LSTM. The research provides valuable insights into Bangla text classification, encouraging further exploration in this area.

Khan et al. (2021) [15] conducted sentiment analysis on Bengali Facebook comments to assess fans' emotions toward celebrities. The dataset comprised 63,000 comments from 12-15 celebrity pages, classified into seven emotion categories: Happy, Sad, Angry, Surprised, Excited, Religious, and Abusive. Various machine learning models were applied, with SVM achieving the highest accuracy of 62%, outperforming Random Forest and K-Nearest Neighbors. The study faced challenges due to an imbalanced dataset, where only 3,000 comments were labeled, affecting overall accuracy.

Bangyal et al. (2021) [16] focused on detecting fake news related to COVID-19 using deep learning techniques. They utilized a dataset of 4,072 articles (2,426 true, 1,646 false) and experimented with various models. Logistic regression achieved 75.65% accuracy, while a stacking-based model attained an F1-score of 0.972. The best result was from a hybrid model combining neural and non-neural features, with a random forest classifier reaching 94.49% accuracy. Limitations include the potential dataset biases and questions about the generalizability of the models to other contexts.

The paper by Bhowmik et al. (2021) [17] develops a domain-specific lexicon data dictionary (LDD) for Bangla sentiment analysis using 50,000 comments from the restaurant and cricket domains, categorized into positive, negative, and neutral classes. They propose a Bangla Text Sentiment Score (BTSC) algorithm and use machine learning methods like SVM, achieving the highest accuracy of 82.21%. The study's limitations include difficulty in detecting neutral sentiments and the time-consuming manual construction of lexicons, affecting scalability and overall accuracy.

Bhowmik et al.(2022) [18] developed a sentiment analysis model for Bengali text using an extended lexicon dictionary and deep learning techniques, achieving an accuracy of 85.8% with a hybrid CNN-LSTM architecture. The model was fine-tuned with a learning rate of 0.0001, a batch size of 25, and 20 epochs. However, limitations include the model's lack of generalization due to a limited dataset, which may introduce bias in the results.

Ghosh et al. (2022) [10] developed a machine learning model to detect depressive social media texts in Bangla, addressing the lack of existing datasets by creating one with 15,031

samples (4,784 depressive and 10,247 non-depressive posts). They proposed an attention-based BiLSTM-CNN model, achieving 94.3% accuracy, with 92.63% sensitivity and 95.12% specificity. The study highlights the need for more diverse datasets and improvements for low-resource languages like Bangla.

Wadud et al. (2022) [19] proposed LSTM-BOOST, an offensive text classification algorithm combining LSTM networks with ensemble learning, to detect offensive Bengali texts on social media. They utilized the Bengali Offensive Text from Social Platforms (BHSSP) dataset with 20,000 posts equally split between offensive and non-offensive content. The model achieved a high F1-score of 92.61%, outperforming other classifiers. However, limitations include difficulty handling long sequence texts and varying accuracy with regional Bengali dialects. Future research aims to address these challenges.

Rahman et al. (2022) [20] focused on sentiment classification of Bengali text using the Word2Vec model, categorizing emotions into happy, angry, and excited. They processed a large dataset, excluding English words, to enhance accuracy. The Skip-Gram model achieved the highest accuracy of 75%, outperforming LSTM and CNN. The study faced challenges with the complexity of the Bengali language and noisy datasets, which impacted the overall performance of sentiment classification.

The paper by Chakraborty et al. (2022) [21] focuses on sentiment analysis of Bengali Facebook data, using a dataset of 10,819 automatically labeled comments and posts, classified into positive and negative sentiments. Seven classifiers were evaluated, including classical methods like Naive Bayes and Random Forest, and deep learning approaches such as LSTM and CNN. The LSTM model achieved the highest accuracy of 96.95%, outperforming the classical models, with Random Forest achieving 78.37%. While limitations like dataset biases are not discussed, the study demonstrates the superior performance of deep learning for Bengali sentiment analysis.

The paper by Prottasha et al. (2022) [22] explores sentiment analysis in the Bangla language using transfer learning with BERT and a hybrid CNN-BiLSTM model. It employs a dataset of 8,952 samples, with 4,325 labeled as positive and the rest negative, sourced from various platforms like social media. The Bangla-BERT model achieved the highest accuracy of 94.15%, outperforming models like Word2Vec and fastText. However, the study noted limitations due to the unbalanced dataset, which may affect the model's efficiency. Future work will focus on expanding the dataset and enhancing real-world performance.

The paper by Aurpa et al. (2022) [23] addresses the detection of abusive Bangla comments on Facebook, using a dataset of 44,001 comments categorized into five classes: sexual, troll, religious, threat, and not bully. Transformer-based deep learning models like BERT and ELECTRA were employed, with BERT achieving the highest test accuracy of 85.00%. The study highlights the lack of labeled datasets for Bangla abusive comments as a limitation, impacting generalizability. This research significantly advances the detection of abusive content in the Bangla language, aiding in better online content moderation.

Rezaul et al. focuses [11] on multi-class sentiment analysis (SA) in the Bengali language, addressing the challenges of classifying sentiments expressed in social media comments. It utilizes a dataset of 42,036 Facebook comments, categorized into four classes: sexual, religious, political, and acceptable. The authors propose a supervised deep learning classifier based on CNN and LSTM architectures, achieving a maximum accuracy of 85.8% and an F1 score of 0.86, which outperforms baseline models. However, the study acknowledges limitations such as the peculiarities of Bengali text, the lack of ground truth datasets, and the scarcity of preprocessing tools, which can hinder performance.

Sourav et al. (2023) [24] developed a transformer-based model for classifying emotions in Bangla texts, using the Unified Bangla Multi-class Emotion Corpus (UBMEC) with 13,436

samples annotated for six emotion classes. The model achieved a weighted F1-score of 71% for the six classes and 76% for a simplified four-class model using m-BERT. Despite its success, the study highlights limitations such as the small dataset size and room for improvement with more training. The paper is a significant contribution to Bangla emotion classification using transformers.

Jahan et al. (2023) [25] developed a method to identify misogynistic content in Bangla on social media, using a dataset of 15,000 comments from Facebook, Instagram, TikTok, and YouTube. The dataset was categorized into misogynistic and non-misogynistic classes, including subcategories like objectification and sexual harassment. The study utilized LSTM and RNN models, with LSTM achieving the highest accuracy of 67%. Limitations include the small dataset size and potential biases from manual labeling. This research contributes to understanding and identifying misogynistic content in Bangla.

The paper by Roy et al. (2023) [26] introduces a new Bengali text classification dataset containing 1,756 documents across 38 classes, such as 'Agriculture,' 'Banking,' and 'Politics.' They evaluated various machine learning and deep learning models, with "FastText with SVC" achieving the highest accuracy of 92.61% and a weighted F1-score of 0.92. The main limitation was confusion between similar classes like "Entertainment" and "Entertainment_other_than_cinema_and_music." This study provides a valuable benchmark for future research in Bengali text classification.

The paper by Wadud et al. (2023) [27] introduces Deep-BERT, a model designed to classify multilingual offensive texts on social media, focusing on English and Bengali. The dataset includes 7000 English and 6500 Bengali comments, categorized into offensive (5085) and non-offensive (8415) classes. The Deep-BERT model achieved the highest accuracy of 93.11%, outperforming other methods like MNB and KNN. The study's limitations include the lack of a standardized multilingual offensive text corpus, which may impact generalizability. This research advances offensive language detection in a multilingual setting using deep learning techniques.

Rahman et al. (2024) [12] conducted sentiment analysis on eLearning texts in Bangla and Romanized Bangla, using three datasets: 3,178 Bangla, 3,090 Romanized Bangla, and 6,268 combined texts, categorized into positive, negative, and neutral classes. They applied machine learning techniques, with XLM-RoBERTa achieving the highest accuracy of 89.46% on the Bangla dataset and 85.81% on the combined dataset, while ANN performed best for Romanized Bangla at 89.59%. The study faced limitations, including cultural context influencing sentiment expression and the need for larger, more diverse datasets. Chowdhury et al.(2024) [38] presents a comprehensive study on detecting depression in Bengali social media texts using a diverse set of models, including deep learning (LSTM, BiLSTM, GRU, BiGRU), transformer models (BERT, BanglaBERT, SahajBERT), and large language models (GPT-3.5, GPT-4, DepGPT). The authors introduce a new Bengali Social Media Depressive Dataset (BSMDD) of 21,910 data of two classes , and demonstrate that their fine-tuned model, DepGPT, outperforms other models with a remarkable F1-score of 0.9804 and accuracy 0.9796. A key strength lies in comparing zero-shot and few-shot learning across models, alongside exploring explainable AI for interpretability. However, the paper lacks a deeper analysis of potential annotation biases, dataset representativeness, and generalizability across different social media platforms. Islam et al. (2024) [39] introduces a large-scale Bangla sentiment analysis dataset (BangDSA) with over 200,000 manually annotated comments across 15 categories, addressing the lack of comprehensive resources for Bangla NLP. It proposes a novel hybrid feature extraction method, skipBangla-BERT, combining Bangla-BERT and Skipgram, which

outperforms traditional approaches. The authors evaluate multiple machine learning, ensemble, and deep learning models, achieving 95.71% accuracy in 3-class sentiment classification using a CNN-BiLSTM architecture. A major strength is the extensive experimentation with 21 feature extraction techniques and statistical validation of results. However, limitations include reliance on manually crawled data from only five platforms, potential annotation biases despite validation, and lack of testing on cross-domain or unseen datasets, affecting generalizability. Future work should explore domain adaptation, transfer learning, and broader linguistic diversity to enhance robustness Mahmud et al. (2024) [40] presents an extensive evaluation of machine learning, deep learning, hybrid, and transformer-based models for multilingual cyberbullying detection in Bangla and Chittagonian texts, introducing a new manually annotated dataset of over 10,000 samples. The authors achieve strong results, with XLM-Roberta reaching 84.1% accuracy and the hybrid (CNN+LSTM)+BiLSTM model attaining 82% accuracy, outperforming traditional approaches. A notable contribution is the inclusion of both Bangla and the low-resource Chittagonian language, along with rigorous annotation validation using Krippendorff's alpha and Cohen's kappa. However, limitations include reliance on a relatively small dataset sourced primarily from a few social media platforms, raising concerns about representativeness and potential domain bias. Future work could expand dataset diversity, assess deployment feasibility, and explore fairness and bias mitigation in multilingual contexts. Rathnayake et al. (2024) [41] showed that socioeconomic indicators can be efficiently modeled using machine learning to comprehend complicated, non-linear interactions that drive gender disparity in Sri Lanka. This encouraged us to reflect on the structured, yet complex patterns inherent in language, such as verb tenses, which often reflect syntactic and morphological dependencies. Despite the fact that we do not deal with socioeconomic factors, the study supported our method of applying machine learning to recognize systematic patterns present in environments with limited resources. The spirit of this study aligns with the objective of our work, which is to use data-driven methods to comprehend and categorize subtle, structurally embedded patterns, in this case, tense variations in Bangla. In a Sri Lankan economic context, Kularathne et al., 2024 [42] shows how machine learning can be used to model real-world phenomena with structured yet variable patterns. Similarly, Bengali verb tense classification requires knowledge of both structured rules (grammar) and exceptions (morphological variability). The combination of machine learning and economic indicators for modeling rice production offered a compelling argument for the use of predictive analytics in low-resource contexts. Bengali NLP faces resource scarcity, much like Sri Lanka's agricultural sector. This study reinforced our decision to rely on models like Random Forest and XGBoost, which have shown success with limited data and complex relationships in time-series or domain-specific forecasting - analogous to how Bengali tense patterns exhibit variability over syntactic and morphological dimensions. The way the authors applied models like XGBoost and decision trees supports our choice to include tree-based classifiers in our ensemble model due to their ability to handle irregular but meaningful patterns in sparse and heterogeneous datasets. Rathnayake et al., 2023 [43] established a strong argument for the use of hybrid learning frameworks and model ensembles, especially in fields where large data variation and noise may make deterministic rule-based modeling ineffective. The fact that individual models would not adequately reflect the grammatical richness of Bangla further strengthened our resolve to combine traditional machine learning with deep learning methods like LSTM and GRU. Their proficiency with ensemble methods and CatBoost verifies our approach of stacking models to increase accuracy and generalizability. This affirmed our design decision to capture the complex structure of Bangla verb forms by merging rule-based and deep learning models. This study [44] suggests a sentiment analysis model that uses ERNIE for encoding and combines contextual semantics and emotional interactions

with BiGRU, attention, and GCN. It outperforms baseline models with a macro-F1 score of 71.90% on the JDDC dataset. However, single-domain data and the lack of multimodal or topic-aware characteristics restrict its use. Table 1 represents the literature summary of related works.

**Table 1. Literature summary of related works.**

| Reference | Algorithm | Classes | Dataset Size | Accuracy | Context |
|---|---|---|---|---|---|
| Kumar Das et al. (2021) [8] | Attention-based Decoder | 2 | 7,425 | 77% | Hate speech detection |
| Asghar et al. (2021) [13] | BiLSTM with Attention | 2 | 11,541 | 92.86% | Sentiment measurement and customer satisfaction prediction |
| Rahman and Chakraborty (2021) [9] | BiLSTM | 12 | 40,000 | 98.87% | Bangla documents classification |
| Salehin et al. (2021) [14] | LSTM, SVM, XGBoost | 12 | 75,951 | 87% | News headline classification |
| Khan et al. 2021) [15] | SVM, Random Forest, KNN | 7 | 63,000 | 62% | Sentiment and emotion classification |
| Bangyal et al. (2021) [16] | Hybrid Neural Model | 1 | 4,072 | 94.49% | Fake news detection |
| Bhowmik et al. (2021) [17] | SVM | 3 | 50,000 | 82.21% | Sentiment analysis |
| Bhowmik et al. (2022) [18] | CNN-LSTM | 3 | 50,000 | 85.8% | Sentiment analysis |
| Ghosh et al. (2022) [10] | BiLSTM-CNN | 2 | 15,031 | 94.3% | Depressive social media text detection |
| Wadud et al. 2022) [19] | LSTM-BOOST | 2 | 20,000 | F1: 92.61% | Offensive text classification |
| Rahman et al. (2022) [20] | Word2Vec, LSTM, CNN | 3 | 11,000 | 75% | Sentiment classification |
| Chakraborty et al. (2022) [21] | LSTM, CNN, Naive Bayes, Random Forest | 2 | 10,819 | 96.95% | Sentiment classification |
| Prottasha et al. (2022) [22] | BERT, CNN-BiLSTM | 2 | 8,952 | 94.15% | Sentiment classification |
| Aurpa et al. (2022) [21 | BERT, ELECTRA | 5 | 44,001 | 85% | Abusive comments detection |
| Rezaul et al. (2023) [11] | CNN, LSTM | 4 | 42,036 | 85.8% | Sentiment analysis |
| Sourav et al. (2023) [24] | (m-BERT) | 6 | 13,436 | 71% | Emotion classification |
| Jahan et al. 2023) [25] | LSTM, RNN | 2 | 15,000 | 67% | Misogynistic content identification |
| Roy et al. (2023) [26] | FastText with SVC | 38 | 1,756 | 92.61% | News headline classification |
| Wadud et al. (2023) [27] | Deep-BERT | 2 | 13,500 | 93.11% | Offensive text classification |
| Rahman et al. (2024) [12] | XLM-RoBERTa, ANN | 3 | 6,268 | 89.59% | Sentiment classification |
| Chowdhury et al.(2024) [38] | LSTM, BiLSTM, GRU, BiGRU, BERT, BanglaBERT, SahajBERT, GPT-3.5, GPT-4, DepGpt | 2 | 21910 | 97.96% | Depression detection |
| Islam et al. (2024) [39] | skipBangla-BERT, CNN-BiLSTM | 15 | 200000 | 95.71% | Sentiment Analysis |
| Mahmud et al. (2024) [40] | SVM, Bangla Bert, M-Bert, Bangla ElectraXLM-Roberta, CNN, LSTM. Bi-LSTM | 4 | 10000 | 84.1% | Multilingual Cyberbullying Detection |
| Rathnayake et al. (2024) [41] | Random Forest (RF), Logistic Regression, Gradient Boosted Trees, XGBoost, Deep Learning Neural Networks | 3 | 21756 | 87.3% | Socioeconomic inequality |
| Kularathne et al., 2024 [42] | Decision Tree (DT), Support Vector Machine (SVM), Random Forest (RF), Gradient Boosted Decision Trees (GBDT),Logistic Regression, LightGBM, CatBoost | 2 | 19116 | 83.25% | Identifying poverty |
| Rathnayake et al., 2023 [43] | Adaptive Neuro-Fuzzy Inference System (ANFIS), Multilayer Perceptron (MLP), Genetic Algorithm (GA) for optimization, Support Vector Regression (SVR), Random Forest (RF) | N/A | 7632 | 96.3% | Forecast air quality |

## Methodology

### Dataset collection and properties

The dataset named "BengaliTenseCorpus: A comprehensive corpus in Bengali texts categorized in Present , Past, and Future" has been sourced from various publicly accessible Bangla blogs, Facebook pages, magazines, books, and news articles, and some of the data are self-made, which ensures a diverse representation of contemporary language use. A critical aspect of the dataset's curation was maintaining an equal distribution of sentences across three tense categories: past, present, and future. The dataset comprises 13,500 Bangla sentences that are categorized into three classes: present tense with 4,550 sentences, past tense with 4,460, and future tense collection with 4,490 sentences (Fig 1). For labeling purposes, 3 numerical values are used as - 0, 1, and 2, respectively, for present tense, past tense, and future tense.

Table 2 represents the statistical summary of the dataset, which is broken down into three different levels: 'Present', 'Past', and 'Future'. Each level summarizes specific statistical measures related to the data distribution. Total number of data points are 4550, 4460, 4490 respectively in 'Present', 'Past' and 'Future' levels. Mean represents the average value of the dataset for each level, 37.75 for Present, 35.03 for Past and 38.63 for Future. The mean and median

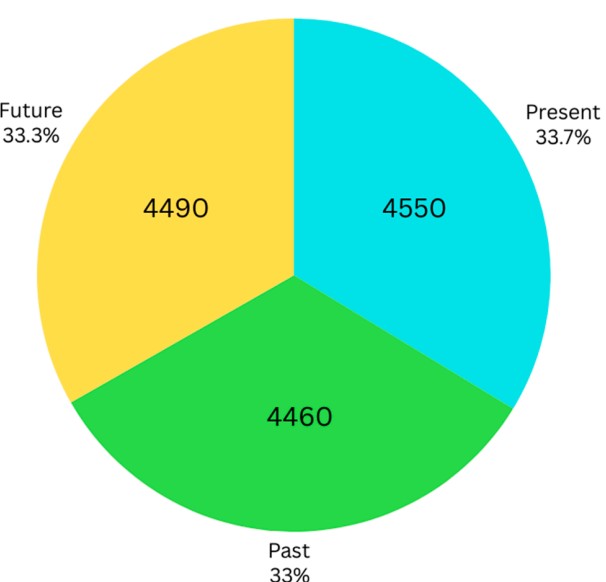

**Fig 1. Statistical analysis of data.**

**Table 2. Detailed statistical summary of dataset.**

| Features | Present | Past | Future |
|---|---|---|---|
| Count | 4,550 | 4,460 | 4,490 |
| Mean | 37.75 | 35.03 | 38.63 |
| Std | 25.55 | 26.00 | 29.45 |
| Min | 9.00 | 9.00 | 10.00 |
| 25% | 21.00 | 22.00 | 19.00 |
| 50% | 27.00 | 27.00 | 25.00 |
| 75% | 50.00 | 35.00 | 52.00 |
| MAX | 202.00 | 262.00 | 216.00 |

values indicate that the 'Future data has a slightly higher average compared to the 'Present' and 'Past' datasets. Similarly, Std shows how much the data varies from the mean (spread or dispersion of the data), Min shows the minimum value in the dataset and we can also find the 25% (1st Quartile), 50% (Median), 75% (3rd Quartile). The standard deviation is highest for the "Future" data (29.45), suggesting greater variability in future data values. Max represents the maximum value present in the dataset for each level. The maximum value in the "Past" category is significantly higher than in the other two categories, which could indicate outliers or larger events in the past.

Fig 2 shows the distribution of text lengths for three classes of data: Present, Past, and Future. The x-axis represents the text length, and the y-axis represents the count (frequency) of texts with that length. For Present Tense, the distribution is heavily skewed to the left, with most text lengths concentrated between 0 and 50 characters. The highest frequency occurs at very short lengths (around 10-25 characters), and the number of texts drastically decreases as the text length increases. The distribution has a long tail, showing that very few texts have lengths above 100 characters. Similar to the present tense, the past tense distribution is also skewed left, but the concentration of text length is even shorter than in the present tense. The highest count is at shorter text lengths, around 10-25 characters, and there are far fewer texts with longer lengths (above 50 characters). The distribution falls off steeply after around 30 characters, suggesting a large number of very short texts. Like the other classes, the future tense distribution is left-skewed, but the data shows a broader spread than the past tense. The concentration is still around shorter text lengths, but more texts appear to have medium lengths (50-75 characters). The highest count is around 20-30 characters, but it gradually decreases compared to the sharp fall in the past tense. The tail is slightly longer than the present and past tense distributions, indicating that the future tense texts tend to have more variability in length. All three classes of tenses show left-skewed distributions, meaning most texts are short in length, though there is some variation, particularly in the future tense category.

Fig 3 represents the Word Cloud for each class of Dataset which visually shows the most frequent words in each dataset, where the size of a word corresponds to its frequency in the dataset. Larger words in the cloud represent higher frequency words for each class.

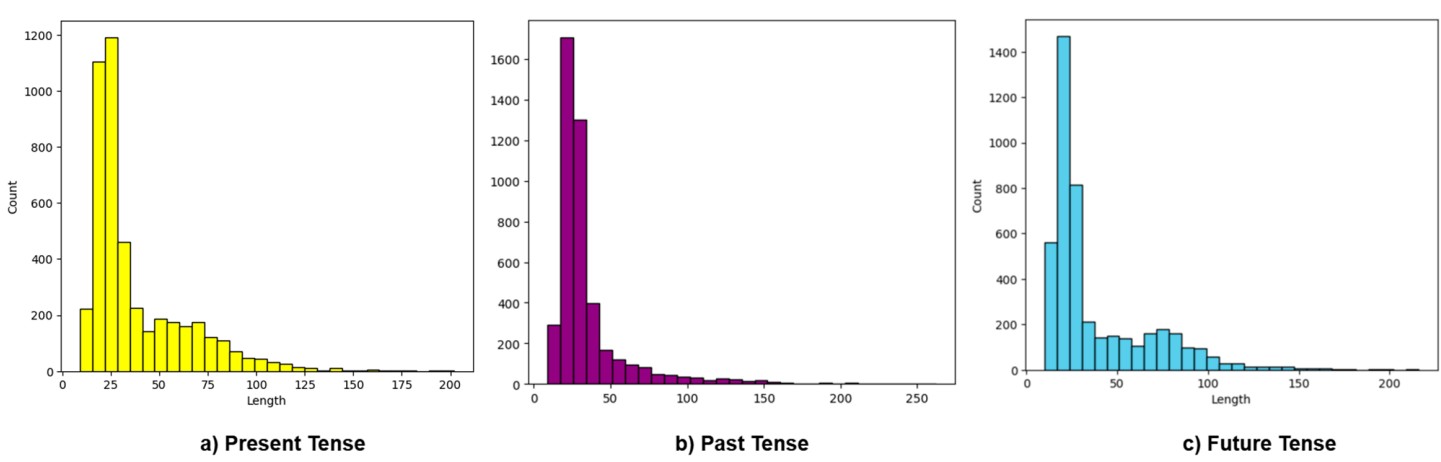

a) Present Tense    b) Past Tense    c) Future Tense

**Fig 2. Distribution of text length of each class of the dataset.**

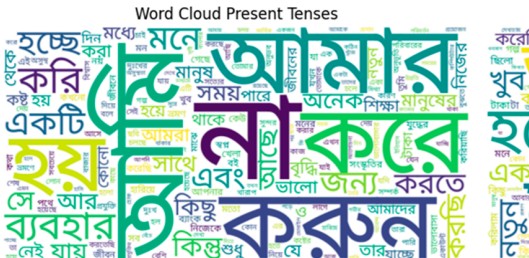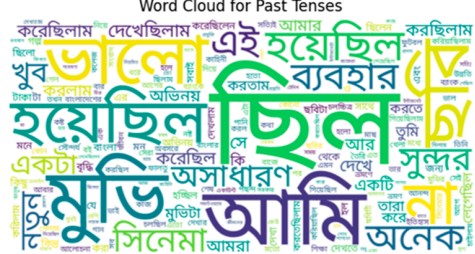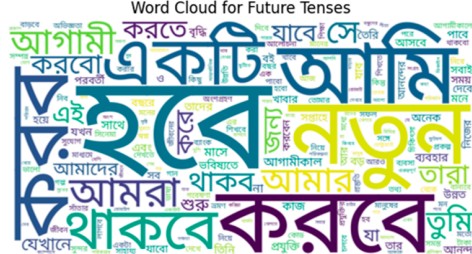

**Fig 3. Word cloud for each class of dataset.**

All relevant data are available from the Mendeley repository at: https://data.mendeley.com/datasets/w9mdy6tw84/1. The data used in this study were collected from publicly available sources in compliance with the terms and conditions of the original providers. All data were anonymized and aggregated where necessary to ensure no violation of privacy or usage restrictions.

## Data pre-processing

Data preprocessing is a very vital fact before starting the classification process. Preprocessing steps are crucial for having an optimal outcome in terms of classification, and outcomes heavily rely on preprocessing. The collected Bengali sentences from diverse sources are not ready to apply machine learning and deep learning approaches without preprocessing, as these are full of noise, lack of coherence, overused punctuation, misspelling, non-standard abbreviations, mixed characters and digits, stop-words, and random use of emojis. Ultimately, classification approaches cannot be applied to these raw data. Thus, it is needed to preprocess the sentences before applying several classification techniques to enhance the overall performances. Several pre-processing steps are applied to our collected data, and these are discussed below as follows:

- **Primary cleaning:** In the primary cleaning step, coherence checking, misspelling checking approaches are applied to initiate the further pre-processing techniques. Coherence ensures the proper structure of a sentence, misspelling checking ensures the correct form of words in the sentences, and null value handling ensures the presentation of all required sentences as well as entails a balanced dataset.
- **Null values Handling:** A small fraction of the rows with null values are identified; remove them from the dataset to ensure quality and integrity of the dataset, reliability and accuracy of the applied classifiers, and computational efficiency.
- **Regular expression removals:** This pre-processing technique is applied to identify and eliminate unexpected patterns or characters, removing punctuation, removing HTML tags and tags, eliminating special characters, discarding unnecessary whitespaces, and eliminating numbers. Eventually, this technique ensures a clean and structured dataset, consistent tokenization, and reduces complexity.
- **Emoji removals:** Emojis are useless according to context; thus, emojis are removed by using replacement methods of Python's emoji package to ensure data consistency and relevant textual content management.
- **Duplicate value removal:** By applying Python's duplicate value handling methods, duplicate values are deleted from the dataset to ensure the uniqueness of the data.

- **Stop word removal:** Stop words are a collection of some unnecessary words that do not carry any meaning in our context. Thus, a list of stop words has been prepared that contains the useless words for this study, and removing this list ensures the minimal training time of models, and improves the performance of the classifiers.
- **Tokenization:** Tokenization, which converts unprocessed text into manageable units (tokens) that machine learning models can analyze, is an essential step in Natural Language Processing (NLP) . It is the basic steps of NLP to simplify the process of models and learn from data, to improve the performance of the classifiers.

Figs 4 and 5 shows the preprocessing steps with examples.

### Data annotation

This study employs manual annotation where human annotators are involved to provide the accurate, relevant, and comprehensive annotations or labels for each entry of the dataset by following guidelines. Three annotators were selected for the accomplishment of the annotation process. The criteria for the annotator selections are that the annotators should be Bengali native speakers as well as have language expertise and linguistic nuances. Actually, the selection criteria included native speakers of Bangla, academic qualifications in Bangla language and linguistics, familiarity with educational content, and ability to discern contexts within this domain. This approach enhances the reliability, validity, and comprehensiveness of tense classification in the Bengali natural language processing domain. Annotators worked individually and independently to avoid bias in labeling or annotating the sentences of the Bengali tense dataset in three forms of tenses: present tense (0), past tense (1), and future (2) by following the Bengali grammatical syntax in terms of tense identification accurately. Table 3 shows the results of annotations as follows: 4550 Bengali sentences are annotated as

**Fig 4. Data sample.**

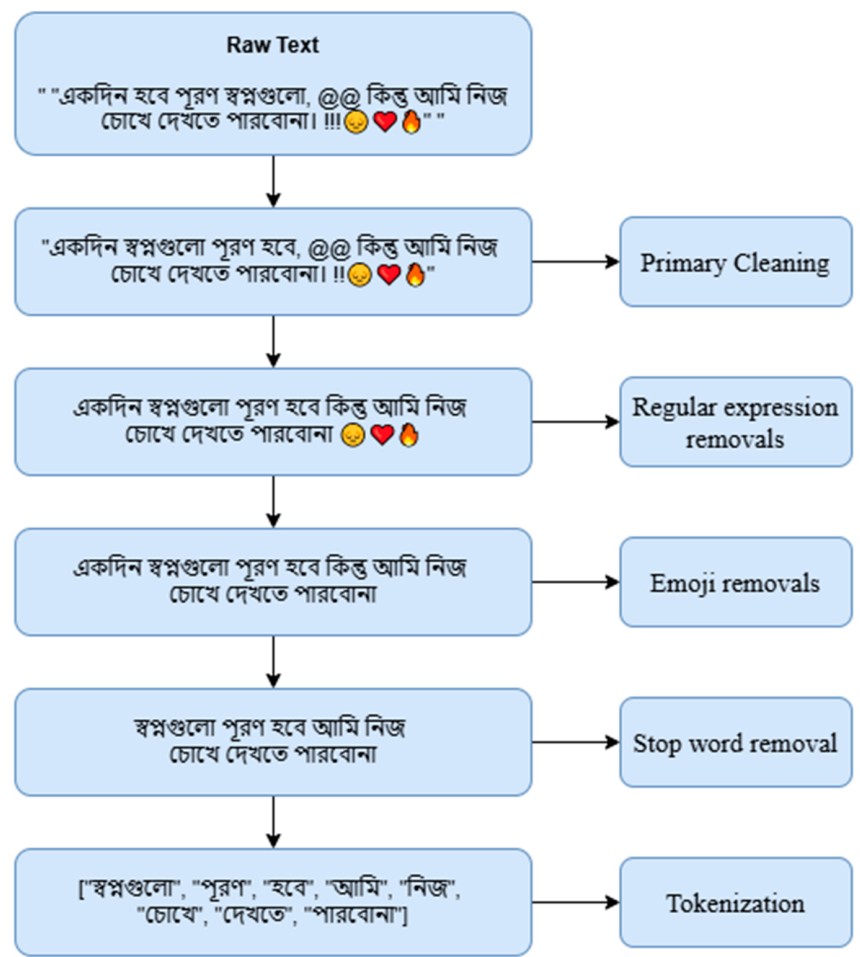

**Fig 5. Data pre-processing techniques.**

**Table 3. Annotation Results.**

| Annotations/Labels | Quantity |
|---|---|
| Present (0) | 4550 |
| Past (1) | 4460 |
| Future (2) | 4490 |

present (0) tense, 4460 sentences are annotated as past (1) tense, and 4490 Bengali sentences are annotated as future (2) tense.

## Model description

**Decision tree.** Decision tree is a commonly used supervised learning method which does not require parameters to do classification tasks. It operates by partitioning the dataset into subsets based on which feature is most informative at each node, creating a tree-like structure of decision rules. The root node represents the entire dataset that is supposed to be splitted into subsets, a decision node or internal node represents a feature, and a leaf node represents an outcome or label. As the splitting criterion, Entropy(Information Gain) is employed to

measure the homogeneity of the nodes and decide the best split at each step of the tree construction. Entropy quantifies uncertainty or impurity in a dataset. In Eq 1, entropy H for a specific dataset T is calculated as:

$$H(T) = -plog_2(p) - (1 - p)log_2(1 - p) \qquad (1)$$

where p is the probability of a sample belonging to the positive class [28].

**Random forest.**   Random Forest is an ensemble learning method that builds multiple decision trees in the training period and afterward combines their output to get a more accurate and stable prediction. Its bootstrapped sampling and random feature selection provide much more diversity among the base estimators, which enhances the robustness of the final model [29]. For classification tasks, the majority vote mechanism is employed by mode which represents the most frequently occurring class among the predictions showing in Eq 2, where if $T_1, T_2, ..., T_M$ are the M trees in Random Forest and $C_i(x)$is the class predicted by the $i^{th}$ tree for input $x$, then the final prediction $\hat{C}(x)$ is determined by:

$$\hat{C}(x) = mode(C_1(x), C_2(x)), ....C_M(x)) \qquad (2)$$

**Multinomial Naive Baye.**   Multinomial Naive Bayes is a probabilistic learning technique that uses Bayes' theorem, assuming independence of features. Despite the simplicity of MNB, it can do surprisingly well even on text datasets with thousands of dimensions. Given the input features, the model determines the posterior probability of each class $P(C_k|x)$ and assigns the class with the highest probability [30].

$$P(C_k|x) = \frac{P(C_k) \prod_{i=1}^{n} P(x_i|C_k)}{P(x)} \qquad (3)$$

In Eq 3, $P(C_k)$ is the prior probability of class $C_k$, $P(x_i|C_k)$ is the likelihood of feature $x_i$ for class $C_k$ and $P(x)$ is the evidence.

**Support vector machine.**   While being a powerful supervised learning algorithm, SVM tends to find the optimal hyperplane that best separates the data points into different classes. This hyperplane maximizes the margin between the two classes; this margin is also referred to as the distance between the closest points of the classes to the hyperplane. Basically, SVM operates nonlinear classification by implicitly mapping the input space to higher dimensions using kernel functions. For a given input vector x, the classification decision is based on the sign of the decision function $f(x)$ given is Eq 4:

$$f(x) = w.x + b \qquad (4)$$

Here, w is the weight vector perpendicular to the hyperplane, x is the input feature vector and b is the bias term [31].

**K Nearest neighbor.**   KNN is an instance-based learning algorithm that classifies a data point by the majority class or most common class of its k nearest neighbors in the feature space. These nearest neighbors are determined based on different distance metrics, among which Euclidean distance is the commonly used one, measured as:

$$d(x, y) = \sqrt{\sum_{i=1}^{n}(x_i - y_i)^2} \qquad (5)$$

In Eq 5, x and y are two points in the n-dimensional feature space [32].

**XGBoost.** XGBoost, or Extreme Gradient Boosting, is an efficient and scalable implementation of gradient boosting for decision trees. It creates a group of weak learners as trees in a sequential manner, each correcting the errors of its predecessor. It also includes a regularization function to prevent overfitting besides other advanced features like tree pruning and parallel processing [33].

$$L = \sum_{i=1}^{n} l(y_i, \hat{y}_i) + \sum_{k=1}^{K} \Omega(f_k) \tag{6}$$

$\sum_{i=1}^{n} l(y_i, \hat{y}_i)$ is the loss function that measures the difference between the predicted value $\hat{y}_i$ and the actual value $y_i$, $\omega$ is the regularization term that penalizes the complexity of the model as shown in Fig 6 and $f_k$ are the individual trees in the Eq 6.

**Recurrent neural networks.** Recurrent neural networks are a class of artificial neural networks meant to recognize patterns in sequences of text. Unlike the traditional feedforward neural networks, in RNNs, connections form directed cycles. An RNN processes data in a sequential manner by keeping some sort of "memory" or hidden state updated at each time step. This hidden state captures information from the past elements of the sequence. It thus allows the network to know the context, and therefore understand how words in a sentence relate to each other. A common architecture of RNN is characterized by the following component shown in Fig 7.

1. **Input Layer** $(x_t)$**:** At each time step $t_1$, the input $x_t$ represents the current word or feature vector in the sequence that is fed into the RNN cell.

2. **Hidden Layer** $(h_t)$**:** The value of the hidden layer is determined based on the present input $x_t$ and the previous hidden state value $h(t-1)$. Mathematically, this may be represented as:

$$h_t = \sigma(W_h x_t + U_h h_{t-1} + b_h) \tag{7}$$

In Eq 7, $W_h$ and $U_h$ are weight metrics, $b_h$ is the bias vector and $\sigma$ is an activation function (generally used tanh or ReLU).

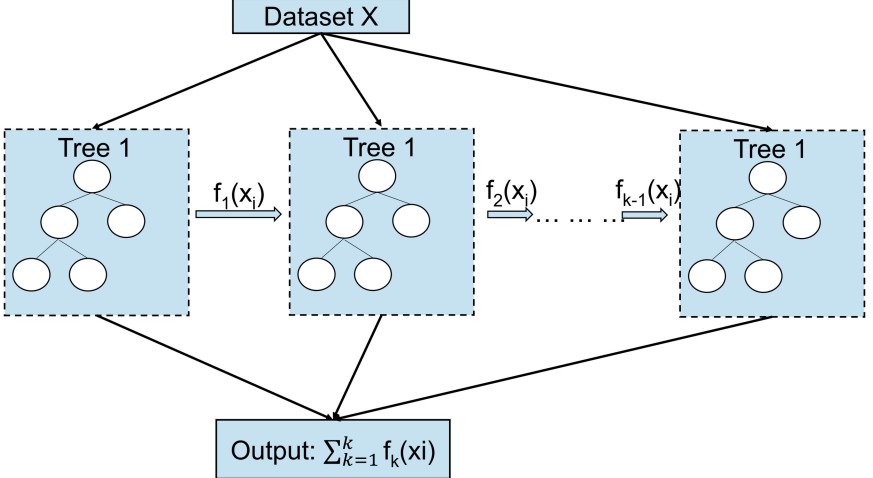

**Fig 6. Work flow of XGBoost.**

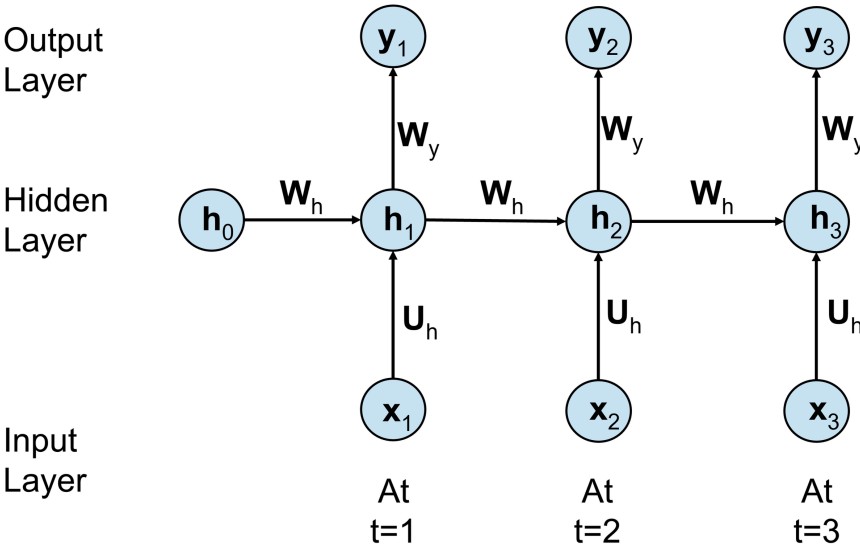

**Fig 7. Simple RNN architecture.**

3.  **Output Layer** ($y_t$)**:** This layer gives the output $y_t$ using the current hidden state $h_t$. Since it's binary classification, mostly it follows through with a sigmoid activation function that gives the probability score.

$$y_t = \sigma(W_y h_t + b_y) \tag{8}$$

In Eq 8, $W_y$ is the output layer weight matrix and usually sigmoid is used as activation function ($\sigma$) to produce a probability between 0 and 1 for binary class [34].

**Long short-term memory.** Long Short-Term Memory networks are a special kind of RNN that is designed to overcome some limitations inherent in traditional RNNs in matters related to handling long-term dependence and how to avoid the problem of vanishing gradients. In LSTMs, this is accomplished by embedding an elaborate architecture composed of memory cells and gates, hence allowing them to keep information for longer sequences. The LSTM networks consist of a series of cells; each cell has three kinds of gates-input gate, forget gate, and output gate. Three kinds of gates are controlling the flow of information inside the cell so that the network should know exactly where to retain or drop off certain information [35]. The architecture of an LSTM cell is illustrated by the following key components portrayed in Fig 8.

1.  **Forget Gate** ($f_t$)**:** It selects which previous cell state information $C(t-1)$ to discard or forget by Eq 9.

$$f_t = \sigma(W_f.[h_{t-1}, x_t] + b_f) \tag{9}$$

2.  **Input Gate** ($i_t$)**:** It calculates what new information coming from the current input $x_t$ needs to be stored in the cell state, combining it with the candidate values ($\hat{C}_t$) using

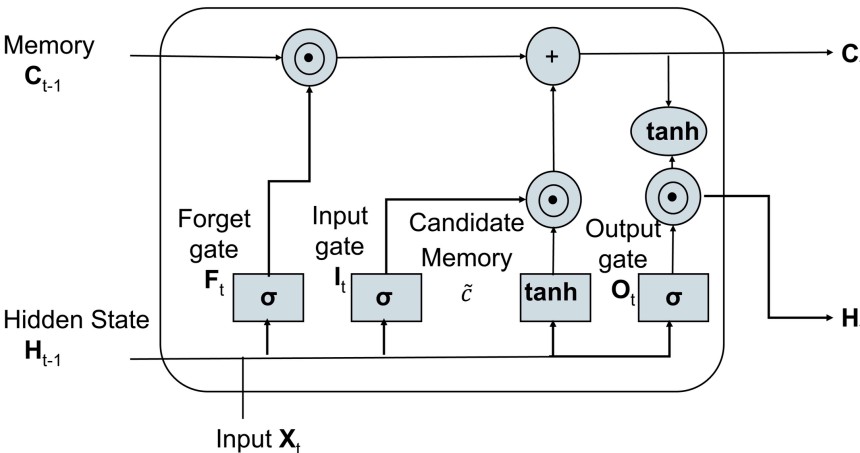

**Fig 8. Internal structure of LSTM.**

Eq 10a and 10b

$$i_t = \sigma(W_i.[h_{t-1}, x_t] + b_i) \tag{10a}$$

$$\hat{C}_t = tanh(W_c.[h_{t-1}, x_t] + b_c) \tag{10b}$$

3. **Cell State** ($C_t$)**:** This is the memory of the network that gets updated at every time step through the combination of the old cell state and the new candidate values, modulated by the forget and input gates, shown in the Eq 11

$$C_t = f_t * C_{t-1} + i_t.\hat{C}_t \tag{11}$$

4. **Output Gate** ($o_t$)**:** This is the gate controlling the cell state output $h_t$ based on the updated cell state $C_t$ and the output gate $o_t$.

$$o_t = \sigma(W_o.[h_{t-1}, x_t] + b_o) \tag{12a}$$

$$h_t = o_t.tanh(C_t) \tag{12b}$$

**Gated recurrent unit.**  Gated Recurrent Unit is a form of RNN, alleviating some issues that used to exist in traditional RNNs; so does the network architecture for LSTMs. They reduce the complexity by fusing the forget and input gates into an update gate, which simplifies the design, decreases computation, and training time, yet yields quality results over sequence data [36]. The major gates in GRUs are the reset gate and the update gate. These two gates help to control the flow of information, which allows the network to capture dependencies in the input sequence. The key components of the GRU cell are visualized in Fig 9,

1. **Reset Gate** ($r_t$)**:** Decides how much of the past state to forget by Eq 13:

$$r_t = \sigma(W_r.[h_{t-1}, x_t] + b_r) \tag{13}$$

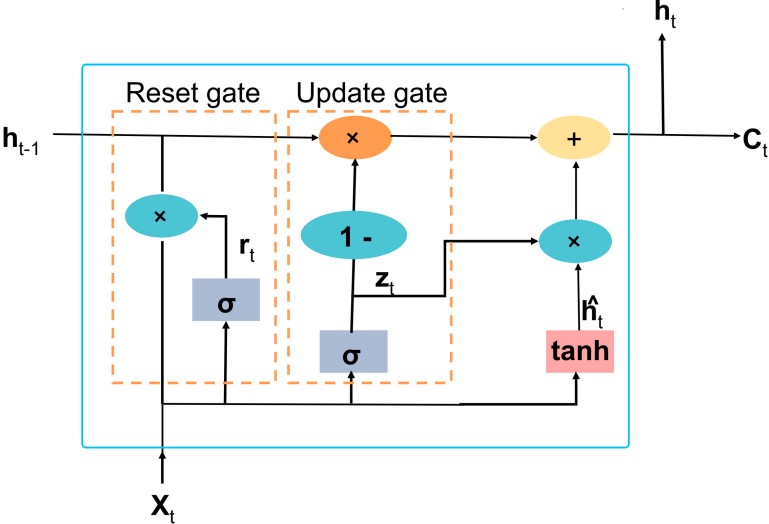

**Fig 9. The architecture of GRU.**

2. **Update Gate** $(z_t)$**:** Updates the balance between the previous state $h(t - 1)$ and the current state $x_t$ by using Eq 14a.

$$z_t = \sigma\big(W_z.[h_{t-1}, x_t] + b_z\big) \tag{14a}$$

$$\hat{h}_t = tanh\big(W.[r_t * h_{t-1}, x_t] + b\big) \tag{14b}$$

$$h_t = (1 - z_t) * h_{t-1} + z_t * \hat{h}_t \tag{14c}$$

Here, Eq 14b calculates the candidate for the new hidden state and the new hidden state $h_t$ is a combination of the old hidden state and the candidate activation, modulated by the update gate represented in Eq 14c [37].

**Stack ensemble model.** Transfer-based models always require a huge dataset to produce acceptable accuracy. Even recurrent neural networks require large datasets to perform fairly. Sometimes, smaller machine learning models can perform as well as sophisticated models. In order to facilitate this quality and boost the performance even more, many tiny models are merged. This type of model is called an ensemble model. Generally, ensemble models are classified into three categories bagging, boosting, and stacking. Because of its durability and ability to learn the final result from the output of previous layers, the stacking ensemble model is frequently chosen over the simple majority voting method. The Stack ensemble model consists of two parts. The first one is generally called the base model, and the second one is called the meta-model. The base model can have several machine learning or deep learning models. The job of the meta-model is to combine the answers of the base model to give the final result. Another key part is to choose which models will be used as base models. This task is often referred to as the model selection part. There is no universal law for choosing models for base models but generally, those models are chosen that are diverse in nature. Often models with greater accuracy can help in overall accuracy. The same analogy goes for the meta-model also. Any machine learning or deep neural network base model can be used as a meta-model.

**Model selection and hyper parameters.** Both machine learning and deep learning models have been used to measure their performance. A total of 9 models have been used for the

model selection part. These models are Naïve Bias, Random Forest, Decision Tree, Support Vector Machine, K-Nearest Neighbor, XGBoost, GRU, and LSTM, and RNN. Multinomial naïve bias is used as it is more suitable for text classification where the parameter alpha is kept at 1.0 and force alpha is True. The following parameters were used to generate the random forest model. The maximum number of trees is 100, and the split criterion is gini. In addition to using the splitter approach 'best', the decision tree classifier's split criterion is also gini. The kernel that is used in the support vector machine is a radial basis function with a degree for the polynomial function is 3. Five neighbors were utilized in the K nearest neighbor model with uniform weights. XGBoost model is generated using all default parameters. One neural network model uses 2 long short-term memory layers one after another. The hidden size of the LSTM was 256, while its input size was 100. After that, it moves through the output layer and three further linear levels. The linear layers have the following architecture (128, 64, 3). With the exception of the output layer, dropout is applied after each linear layer. The dropout value was 0.4, and the in-place option was set to false. The learning rate was 0.001, the batch size was equal to 10, and there were 100 epochs in total. The best weights were recorded while training. The gated recurrent unit was employed by another neural network. Five linear layers were traversed after using two gated recurrent units consecutively. GRU had a hidden size of 256 and an input size of 100. The output layer is the final of the five linear layers. The linear layers' architecture, which matches the lstm's layers, was (128, 64, 32, 16, 3). There are 100 epochs, a batch size of 10, and a learning rate of 0.001. With the exception of the output layer, dropout is applied after each linear layer. The dropout value was 0.4, and the in-place option was set to false. The models' optimal weights were also recorded. The last neural network model was RNN. The model architecture of the RNN model was as follows. Two RNN layers were stacked one after another. The input size for the RNN was 100 and the hidden size was 256. Then a flatten layer is used before passing them to linear layers. 3 linear layers (128, 64, 3) were used with dropout layers in between them except for the final output layer. The dropout rate was 40% with the inplace option being false. The number of epochs was 100 with a learning rate of 0.001 and batch size being 100.

**Our model.**   A total of 5 models were selected based on their performance to use in our ensemble model. These models were the Random Forest Model, Support Vector Machine, XGBoost classifier, Long Short-Term Memory (LSTM), and Gated Recurrent Unit (GRU). At the time of model selection part all models were saved. Each model demonstrated strong predictive ability individually, which makes them suitable candidates for contributing diverse and accurate predictions in the ensemble. Our ensemble model consists of machine learning models and deep learning models. These models use different learning biases. This diversity is essential in stacking to ensure the meta-learner can generalize well across prediction errors. In our base model, we utilized these saved models. A neural network is employed for the meta-model, which will integrate the performance of basic models. The neural network consists of 5 layers input layer, hidden layers, and output layer. The size of the input layer is 5 because of the five base models. The output of the five base models will be used as input for the input layer. The number of nodes in the hidden layer was kept at (128, 64, 32, 3) and for the output layer, it was 3 representing three of our labels representing present, past, and future. In training the meta model base models were kept frozen. The input is presented to the base models to predict the output. The predicted output then was used as input for the meta-model and the meta model will be trained by updating its parameters based on its prediction and actual label. The output of the meta model is considered to be the final output. The learning rate for our meta model was 0.001 with 100 epochs for training and the batch size was 100 with regularization r2 = 0.001. Based on train and test accuracy the weights of the best model were stored. The overall architecture of our model can be seen in the Fig 10.

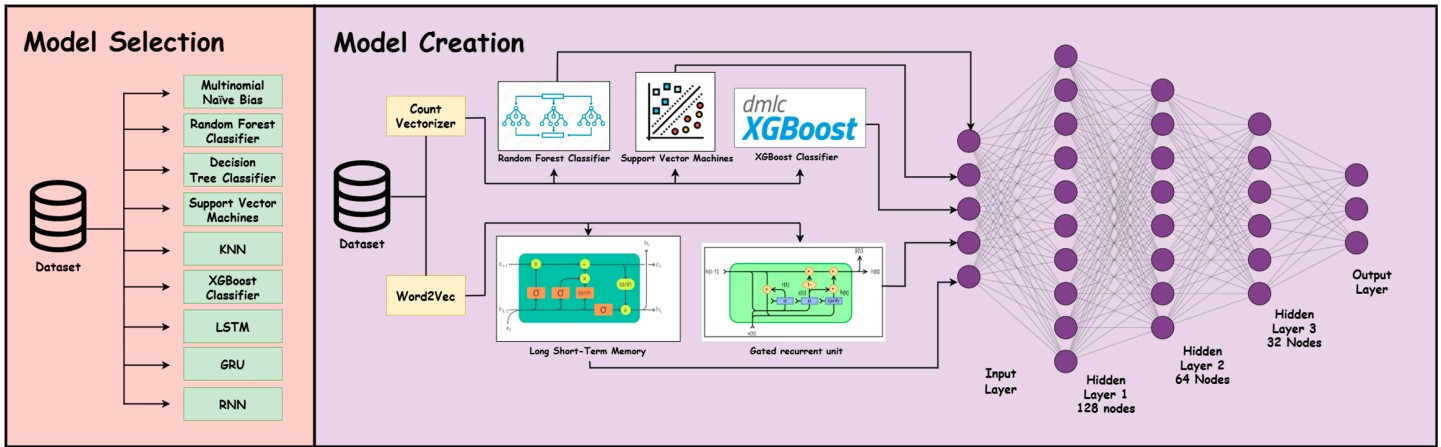

**Fig 10. Model architecture.**

### Use of generative AI

The authors have employed generative AI to help with manuscript writing, however, it was not utilized to create the source code or while creating the dataset.

- Grammarly: To fix grammatical errors Grammarly software was used. The authors double-checked the suggestions given by the software whether the suggestions would alter the semantic text.
- QuillBot: Some texts were paraphrased using the QuillBot program in the literature review and introduction part and the writers verified the results. We have used QuillBot to paraphrase our own texts. We didn't upload published works to the tool to produce the literature review. The QuillBot app is only used to reword content in a more appropriate format. It is not used to generate text for introductions or literary reviews.

## Results and discussion

This section states the evaluation result of the proposed combined model consisting of machine learning models Random Forest, Support Vector Machine, XGBoost with deep learning models GRU, LSTM on our Bangla Tense dataset. Initially, nine individual experiments of machine learning and deep learning models were conducted to analyze their performance on our dataset and the models are Decision Tree, KNN, Multinomial Naive bayes, Random Forest, RNN, SVM, XGBoost, GRU, LSTM. After that, the five models were chosen to combine based on their performance. Table 4 shows the individual experimental results which were gained by choosing the optimal values for each parameter through necessary trial and error.

### Machine learning models

Fig 11 shows the confusion matrices for the DT, KNN, MNB, RF, SVM, XGB classifiers for our dataset. The confusion matrix for decision tree in Fig 11(a) shows that 314 instances of class 0 (Present) were correctly classified as 0, but 73 were misclassified as 1 (Past), and 68 were misclassified as 2 (Future). 337 instances of class 1 were correctly classified, but 87 were predicted as 0, and 22 as 2. 355 instances of class 2 were correctly classified, but 71 were predicted as 0, and 23 as 1. The confusion matrix for KNN in Fig 11(b) shows that 285 instances

**Table 4. Performance analysis of machine learning models.**

| Model | Level | Precision | Recall | F1-score | Support |
|---|---|---|---|---|---|
| Decision tree | 0 (Present) | 0.67 | 0.69 | 0.68 | 455 |
| | 1 (Past) | 0.78 | 0.76 | 0.77 | 446 |
| | 2 (Future) | 0.80 | 0.79 | 0.79 | 449 |
| K neighbours | 0 (Present) | 0.46 | 0.63 | 0.53 | 455 |
| | 1 (Past) | 0.63 | 0.51 | 0.56 | 446 |
| | 2 (Future) | 0.67 | 0.56 | 0.61 | 449 |
| Multinomial naive bayes | 0 (Present) | 0.59 | 0.56 | 0.57 | 455 |
| | 1 (Past) | 0.66 | 0.68 | 0.67 | 446 |
| | 2 (Future) | 0.75 | 0.78 | 0.76 | 449 |
| Random forest | 0 (Present) | 0.78 | 0.63 | 0.70 | 455 |
| | 1 (Past) | 0.81 | 0.85 | 0.83 | 446 |
| | 2 (Future) | 0.78 | 0.89 | 0.83 | 449 |
| SVM | 0 (Present) | 0.71 | 0.65 | 0.68 | 455 |
| | 1 (Past) | 0.78 | 0.78 | 0.78 | 446 |
| | 2 (Future) | 0.76 | 0.82 | 0.79 | 449 |
| XGBoost | 0 (Present) | 0.85 | 0.47 | 0.61 | 455 |
| | 1 (Past) | 0.75 | 0.83 | 0.79 | 446 |
| | 2 (Future) | 0.70 | 0.94 | 0.80 | 449 |

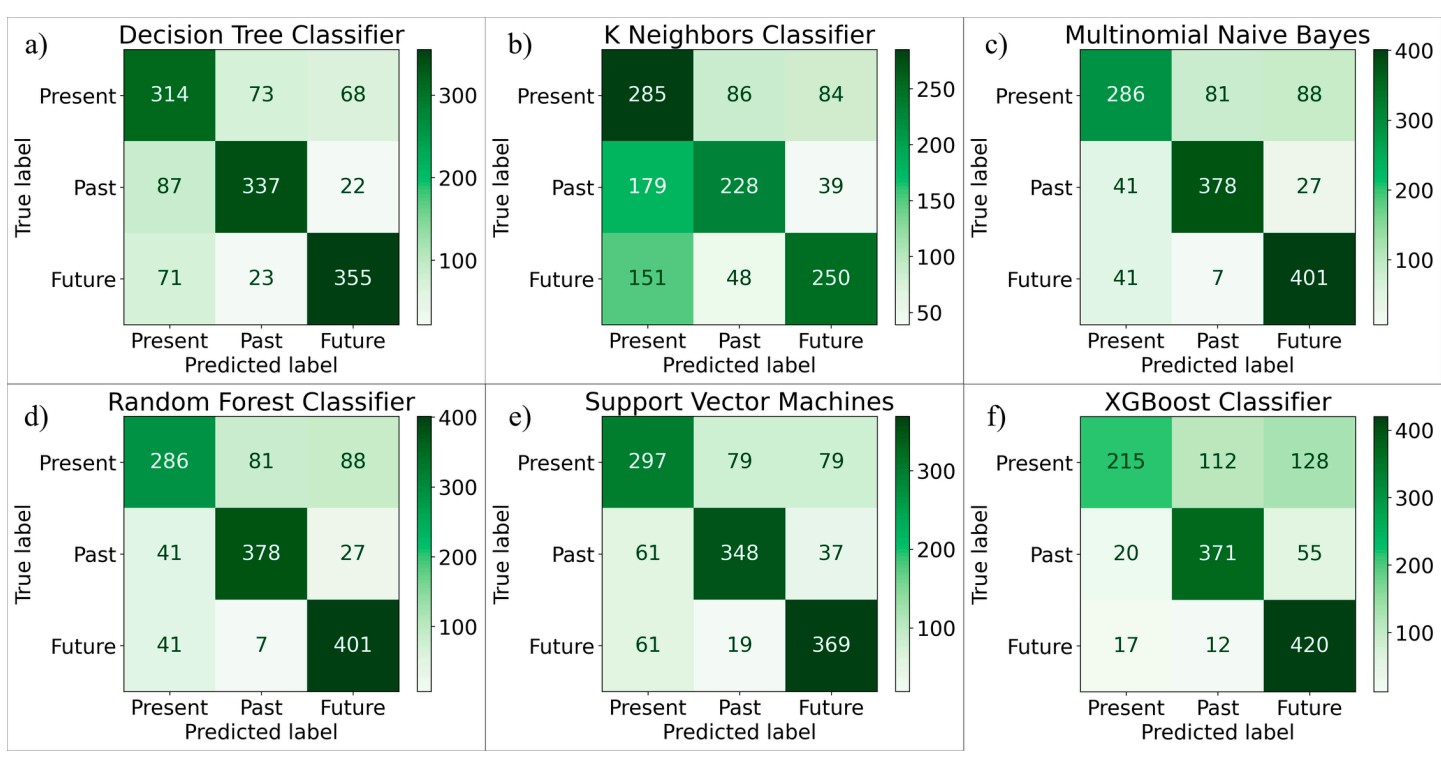

**Fig 11. Confusion matrices for machine learning models (a) DT(b) KNN (c) MNB (d) RF (e) SVM (f) XGB.**

of class 0 (Present) were correctly classified as 0, but 86 were misclassified as 1 (Past), and 84 were misclassified as 2 (Future). 228 instances of class 1 were correctly classified, but 179 were predicted as 0, and 39 as 2. 250 instances of class 2 were correctly classified, but 151

were predicted as 0, and 48 as 1. The confusion matrix for the multinomial naive bayes classifier in Fig 11(c) shows that 286 instances of class 0 (Present) were correctly classified as 0, but 81 were misclassified as 1 (Past), and 88 were misclassified as 2 (Future). 378 instances of class 1 were correctly classified, but 41 were predicted as 0, and 27 as 2. 401 instances of class 2 were correctly classified, but 41 were predicted as 0, and 7 as 1. The confusion matrix for the random forest classifier in Fig 11(d) shows that 286 instances of class 0 (Present) were correctly classified as 0, but 81 were misclassified as 1 (Past), and 88 were misclassified as 2 (Future). 378 instances of class 1 were correctly classified, but 41 were predicted as 0, and 27 as 2. 401 instances of class 2 were correctly classified, but 41 were predicted as 0, and 7 as 1. The confusion matrix for the svm based classifier in Fig 11(e) shows that 297 instances of class 0 (Present) were correctly classified as 0, but 79 were misclassified as 1 (Past), and 79 were misclassified as 2 (Future). 348 instances of class 1 were correctly classified, but 61 were predicted as 0, and 37 as 2. 368 instances of class 2 were correctly classified, but 61 were predicted as 0, and 19 as 1. The confusion matrix for the xgboost classifier in Fig 11(f) shows that 215 instances of class 0 (Present) were correctly classified as 0, but 112 were misclassified as 1 (Past), and 128 were misclassified as 2 (Future). 371 instances of class 1 were correctly classified, but 20 were predicted as 0, and 55 as 2. 420 instances of class 2 were correctly classified, but 17 were predicted as 0, and 12 as 1.

As per the Table 4, for Decision Tree(DT) ,67% were correctly predicted and 69% were correctly identified as class 0 among 455 actual instances of class 0, 78% were correctly predicted and 76% were correctly identified as class 0 among 446 actual instances of class 1, 80% were correctly predicted and 79% were correctly identified as class 2 among 449 actual instances of class 2. The model performs best for class 2, as it has the highest precision (80%) and recall (79%) compared to the other classes. This indicates that class 2 predictions are the most accurate, and the model can correctly identify the majority of class 2 instances. For KNN, 46% were correctly predicted and 63% were correctly identified as class 0 among 455 actual instances of class 0, 63% were correctly predicted and 51% were correctly identified as class 0 among 446 actual instances of class 1, 67% were correctly predicted and 56% were correctly identified as class 2 among 449 actual instances of class 2. It performs better for class 2 compared to classes 0 and 1, based on precision, recall, and F1-scores. For Multinomial Naive Bayes (MNB), 59% were correctly predicted and 56% were correctly identified as class 0 among 455 actual instances of class 0, 66% were correctly predicted and 68% were correctly identified as class 0 among 446 actual instances of class 1, 75% were correctly predicted and 78% were correctly identified as class 2 among 449 actual instances of class 2. It performs best for class 2 compared to classes 0 and 1, based on precision, recall, and F1-scores. For Random Forest (RF), 78% were correctly predicted and 63% were correctly identified as class 0 among 455 actual instances of class 0, 81% were correctly predicted and 85% were correctly identified as class 0 among 446 actual instances of class 1, 78% were correctly predicted and 89% were correctly identified as class 2 among 449 actual instances of class 2. It performs better for class 1 compared to classes 0 and 2, based on precision, recall, and F1-scores. For Support Vector Machines (SVM), 71% were correctly predicted and 65% were correctly identified as class 0 among 455 actual instances of class 0, 78% were correctly predicted and 78% were correctly identified as class 0 among 446 actual instances of class 1, 76% were correctly predicted and 82% were correctly identified as class 2 among 449 actual instances of class 2. The model performs better for class 1, as it achieves a high and balanced precision and recall, making it more reliable in both predicting and identifying class 1 instances. For Xgboost (Xgb), 85% were correctly predicted and 47% were correctly identified as class 0 among 455 actual instances of class 0, 75% were correctly predicted and 83% were correctly identified as class 0 among

446 actual instances of class 1, 70% were correctly predicted and 94% were correctly identified as class 2 among 449 actual instances of class 2. Considering precision and recall together, class 2 performs better overall due to its higher precision (94%) and reasonably good recall (70%). While class 0 has the highest recall, its much lower precision (47%) indicates more false positives, reducing its overall performance compared to class 2.

## Deep learning models

Fig 12(a) shows the confusion matrix for the GRU based classifier for the training data. Fig 12(b) shows the confusion matrix for the LSTM based classifier for the training data. Fig 12(c) shows the confusion matrix for the RNN based classifier for the training data. Fig 13(a) shows the confusion matrix for the GRU for the test data. 299 instances of class 0 (Present) were correctly classified as 0, but 137 were misclassified as 1 (Past), and 19 were misclassified as 2 (Future). 355 instances of class 1 were correctly classified, but 70 were predicted as 0, and 21 as 2. 385 instances of class 2 were correctly classified, but 42 were predicted as 0, and 22 as 1. Fig 13(b) shows the confusion matrix for the LSTM for the test data. 344 instances of class 0 (Present) were correctly classified as 0, but 82 were misclassified as 1 (Past), and 29 were misclassified as 2 (Future). 326 instances of class 1 were correctly classified, but 101 were predicted as 0, and 19 as 2. 377 instances of class 2 were correctly classified, but 37 were predicted as 0, and 35 as 1. Fig 13(c) shows the confusion matrix for the RNN

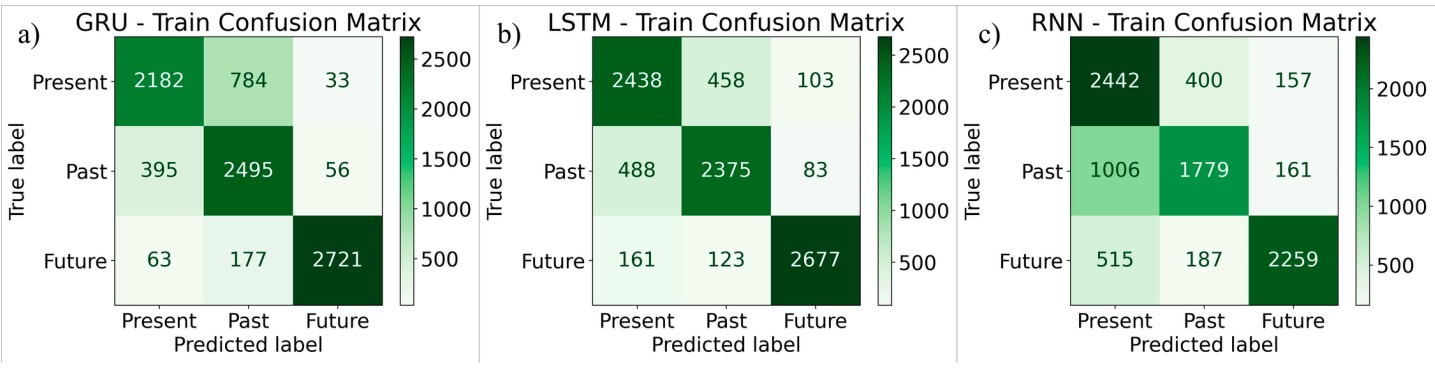

**Fig 12. Confusion matrices for training data for deep learning models (a) GRU classifier (b) LSTM classifier (c) RNN classifier.**

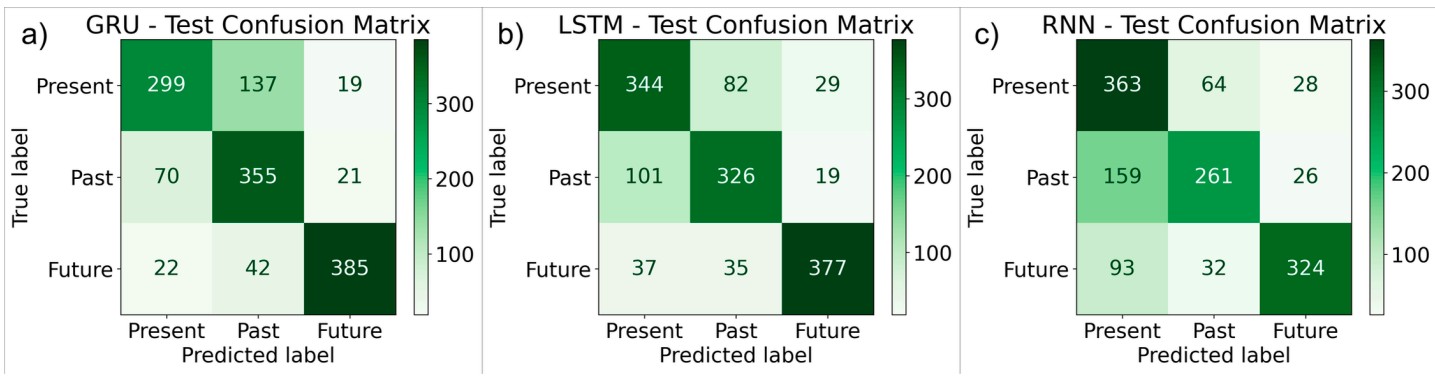

**Fig 13. Confusion matrices for test data for deep learning models (a) GRU classifier (b) LSTM classifier (c) RNN classifier.**

classifier for the test data. 363 instances of class 0 (Present) were correctly classified as 0, but 64 were misclassified as 1 (Past), and 28 were misclassified as 2 (Future). 261 instances of class 1 were correctly classified, but 159 were predicted as 0, and 26 as 2. 324 instances of class 2 were correctly classified, but 93 were predicted as 0, and 32 as 1.

Table 5 represents the performance analysis of GRU, LSTM, and RNN on training data. With a precision of 0.97, recall of 0.92, and F1-score of 0.94, the GRU model does quite well when it comes to future tense prediction. Additionally, it has a good recall of 0.85 and an F1-score of 0.78 when it comes to the past tense. Its accuracy in the present tense, however, is marginally lower at 0.83, yielding an F1-score of 0.77. With F1-scores for Present and Past close to 0.80 and for Future close to 0.92, the LSTM model performs well in all tenses. The RNN model trails behind GRU and LSTM in terms of present tense, future tense, and past tense prediction accuracy. Table 6 illustrates the performance analysis of GRU, LSTM, and RNN on the test data. Where GRU has the highest performance in predicting Future tense with a precision of 0.91, recall of 0.86, and an F1-score of 0.88. Its performance in Present tense is lower, with a precision of 0.76 and an F1-score of 0.71, suggesting it struggles more with differentiating present tense sentences. The Past tense performs moderately well, with an F1-score of 0.72 but slightly higher recall (0.80), suggesting it retrieves more correct past tense sentences but with less precision (0.66). LSTM stands out for its balanced performance across all tenses, especially for the Present and Past tenses. It achieves precision of 0.76 and 0.77, with corresponding F1-scores of 0.76 and 0.77, showing strong capability in maintaining a good balance between precision and recall for both. RNN shows weaker performance in tense classification, with a precision of 0.59 in Present tense and a strong recall of 0.80. Future tense predictions for RNN are still relatively good with an F1-score of 0.78, though not as strong as LSTM or GRU. The average accuracy of GRU, LSTM and RNN was 77.11%, 77.48%, and 70.67%.

**Table 5. Performance analysis of Deep learning models (Training Data).**

| Models | Level | Precision | Recall | F1-score | Support |
|---|---|---|---|---|---|
| GRU | 0 (Present) | 0.83 | 0.73 | 0.77 | 2999 |
| | 1 (Past) | 0.72 | 0.85 | 0.78 | 2946 |
| | 2 (Future) | 0.97 | 0.92 | 0.94 | 2961 |
| LSTM | 0 (Present) | 0.79 | 0.81 | 0.80 | 2999 |
| | 1 (Past) | 0.80 | 0.81 | 0.80 | 2946 |
| | 2 (Future) | 0.94 | 0.90 | 0.92 | 2961 |
| RNN | 0 (Present) | 0.62 | 0.81 | 0.70 | 2999 |
| | 1 (Past) | 0.75 | 0.60 | 0.67 | 2946 |
| | 2 (Future) | 0.88 | 0.76 | 0.82 | 2961 |

**Table 6. Classification report for all deep learning models on (Test Data).**

| Models | Level | Precision | Recall | F1-score | Support |
|---|---|---|---|---|---|
| GRU | 0 (Present) | 0.76 | 0.66 | 0.71 | 455 |
| | 1 (Past) | 0.66 | 0.80 | 0.72 | 446 |
| | 2 (Future) | 0.91 | 0.86 | 0.88 | 449 |
| LSTM | 0 (Present) | 0.76 | 0.75 | 0.76 | 455 |
| | 1 (Past) | 0.77 | 0.76 | 0.77 | 446 |
| | 2 (Future) | 0.84 | 0.87 | 0.86 | 449 |
| RNN | 0 (Present) | 0.59 | 0.80 | 0.68 | 455 |
| | 1 (Past) | 0.73 | 0.59 | 0.65 | 446 |
| | 2 (Future) | 0.86 | 0.72 | 0.78 | 449 |

## Proposed combined model

Random Forest, Support Vector Machine, XGBoost, LSTM, GRU models are combined together as the proposed model with specific parameters. Fig 14(a) shows the confusion matrix for the proposed combined classifier for the training data. Fig 14(b) shows the confusion matrix for the proposed combined model classifier for the test data. 335 instances of class 0 (Present) were correctly classified as 0, but 61 were misclassified as 1 (Past), and 59 were misclassified as 2 (Future). 394 instances of class 1 were correctly classified, but 29 were predicted as 0, and 23 as 2. 410 instances of class 2 were correctly classified, but 29 were predicted as 0, and 10 as 1.

Table 7 represents the performance analysis of the proposed combined model for the training data and test data separately. The model shows near-perfect classification performance on the train data, achieving outstanding results across all tenses with a Precision, Recall, and F1-score around 0.95-0.96 for the Present, Past, and Future categories. But performance falters on test data, especially with the Present and Future tenses. The Present tense shows lesser precision (0.87) and F1-score (0.79), implying greater misclassifications on unseen data, but the Past tense still performs reasonably well with an F1-score of 0.86. A decline in performance is also seen in the Future tense, with an F1-score of 0.87. This suggests that while the model generalizes effectively, testing on new data presents certain difficulties in sustaining the same degree of accuracy. The average accuracy was 85.34%.

Fig 15 shows the accuracies of the used machine learning classifiers (Naive Bayes, Random Forest, Decision Tree, Support Vector Machine, K Nearest Neighbor, XGBoost) for the test data which are 67.037%, 78.8889%, 74.5185%, 75.1111%, 56.5185% and 74.5185% respectively. On the other hand, the training accuracy is 82.596%, test accuracy is 77.11% for the GRU based model. The training accuracy is 83.45.%, test accuracy is 77.48% for the

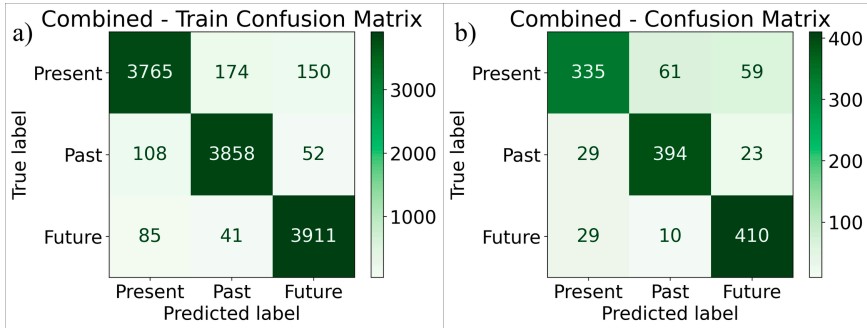

**Fig 14. Confusion matrix for proposed combined classifier (a) training data (b) test data.**

**Table 7. Performance analysis of the proposed combined on train data and test data.**

| Dataset | Level | Precision | Recall | F1-score | Support |
|---|---|---|---|---|---|
| Train Data | 0 (Present) | 0.95 | 0.92 | 0.94 | 2999 |
| | 1 (Past) | 0.95 | 0.96 | 0.95 | 2946 |
| | 2 (Future) | 0.95 | 0.97 | 0.96 | 2961 |
| Test Data | 0 (Present) | 0.87 | 0.73 | 0.79 | 455 |
| | 1 (Past) | 0.83 | 0.89 | 0.86 | 446 |
| | 2 (Future) | 0.83 | 0.92 | 0.87 | 449 |

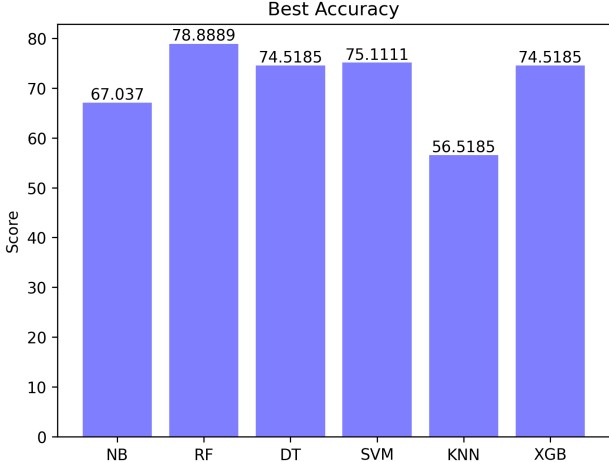

**Fig 15. Accuracy of existing machine learning models.**

LSTM model. The training accuracy is 73.669% and test accuracy is 70.67% for the RNN model.

Finally, for the proposed combined model shown in Fig 16(a) and 16(b) depicts the accuracy and loss curve of the proposed combined model over 20 epochs. The model's test accuracy peaks at 85.34%, while the training accuracy rises to 95%. This suggests the model is learning well but may need more fine-tuning on untested data. The training loss decreases, indicating the model is reducing errors on the training set. The test loss remains flat, suggesting improved generalization and mild overfitting, where the model performs better on training data than test data. In Fig 16(c), The model's performance is evaluated for three classes on the ROC curve: negative, positive, and neutral, each with an AUC score. The model is more effective at differentiating between classes, with the Neutral class having the highest AUC (0.97). The Positive class follows with an AUC of 0.95, and the Negative class has the lowest

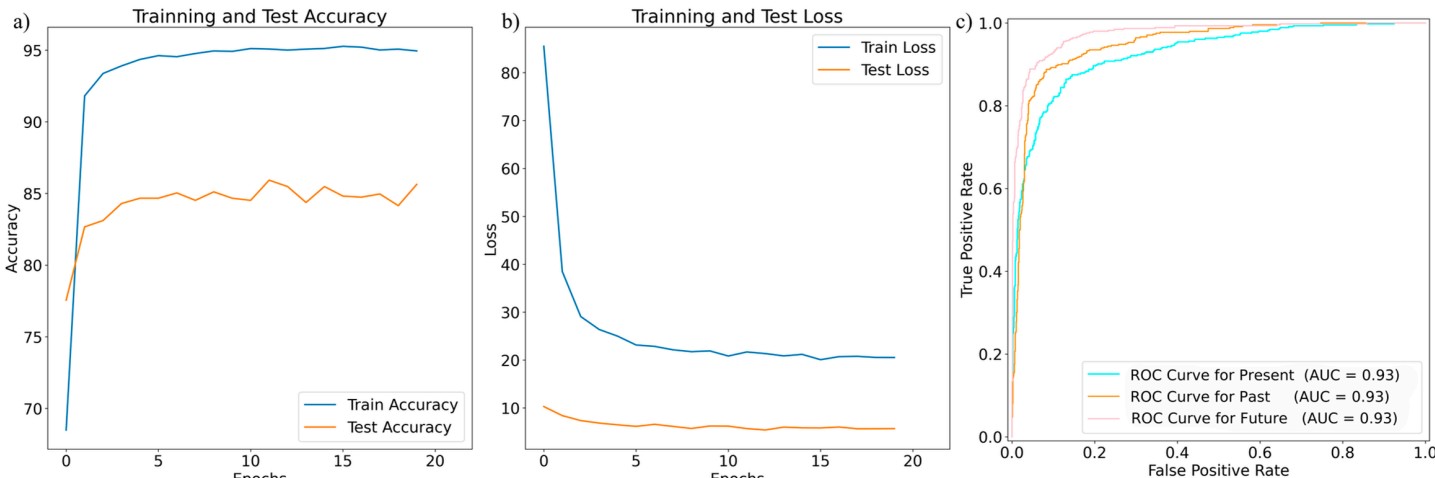

**Fig 16. Proposed combined model (a) training and test accuracy (b) training and test loss (c) ROC curve of proposed combined model.**

but still significant AUC of 0.93. The model's excellent predictive ability is demonstrated in all classes, with Neutral forecasts showing particularly impressive performance.

## Conclusion

In order to meet a critical requirement in natural language processing (NLP) applications for one of the most widely spoken languages in the world, this work presents a novel stacked ensemble model for Bengali tense prediction. The study demonstrates the potential of ensemble approaches in resolving challenging language problems by utilizing the recently engendered BengaliTense dataset, which consists of 13,500 annotated phrases divided into three classes: Present, Past, and Future. To ensure excellent data quality and dependability for experimentation, the dataset was painstakingly preprocessed using methods including contextual annotation, noise removal, and missing data handling. The strengths of several classifiers, including XGBoost, Random Forest, SVM, GRU, and LSTM, are combined in the suggested ensemble model to create a strong framework. By achieving an accuracy of 85% on the test set, this hybrid strategy outperformed standalone models. The ensemble's practical usefulness in real-world applications is shown by its ability to handle Bengali's morphological richness and tense-specific changes. Additionally, comparison analyses provided information about the particular advantages and disadvantages of each model, demonstrating the ensemble's superior generalization across a range of linguistic structures.

While the results are promising, some limitations were identified, including challenges with unseen data and the need for larger and more diverse datasets. Future research should focus on expanding the dataset, incorporating advanced models such as transformers, and exploring morphological tense conversion for a comprehensive approach to Bengali NLP. The proposed ensemble model performs well but shows limited generalization on unseen data, and the dataset may not capture all Bengali dialects. Transformer-based models were not explored, and standardized benchmarks are lacking. Future work will address these issues by expanding the dataset, integrating advanced models, and exploring tense morphology. In future work, we aim to incorporate token-level attribution methods such as SHAP analysis.

This work lays a strong foundation for subsequent studies in the field and contributes significantly to the development of inclusive language technologies.

## Acknowledgments

Students at Daffodil International University are appreciated for their assistance in creating the dataset.

## Author contributions

**Data curation:** Umme Ayman, Ms. Nusrat Khan, Ms. Chayti Saha.

**Formal analysis:** Md. Nahid Hasan, Md. Fayejullah.

**Methodology:** Md. Nahid Hasan.

**Supervision:** Md. Nahid Hasan.

**Visualization:** Md. Nahid Hasan.

**Writing – original draft:** Umme Ayman.

**Writing – review & editing:** Umme Ayman, Ms. Nusrat Khan, Ms. Chayti Saha, Md. Fayejullah.

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
