## [Decision Letter · Decision Letter 0]

9 Apr 2025

PONE-D-24-57203Navigating Tenses in Bengali Sentences: A Stacked Ensemble Model for Enhanced PredictionPLOS ONE

Dear Dr. Hasan,

Thank you for submitting your manuscript to PLOS ONE. After careful consideration, we feel that it has merit but does not fully meet PLOS ONE’s publication criteria as it currently stands. Therefore, we invite you to submit a revised version of the manuscript that addresses the points raised during the review process.

We look forward to receiving your revised manuscript.

Kind regards,

Namal Rathnayake, Ph.D.

Academic Editor

PLOS ONE

**Journal Requirements:**

Please ensure that your manuscript meets PLOS ONE's style requirements, including those for file naming. The PLOS ONE style templates can be found at https://journals.plos.org/plosone/s/file?id=wjVg/PLOSOne_formatting_sample_main_body.pdf and https://journals.plos.org/plosone/s/file?id=ba62/PLOSOne_formatting_sample_title_authors_affiliations.pdf 2. In your Methods section, please include additional information about your dataset and ensure that you have included a statement specifying whether the collection and analysis method complied with the terms and conditions for the source of the data. 3. Please note that PLOS ONE has specific guidelines on code sharing for submissions in which author-generated code underpins the findings in the manuscript. In these cases, we expect all author-generated code to be made available without restrictions upon publication of the work. Please review our guidelines at https://journals.plos.org/plosone/s/materials-and-software-sharing#loc-sharing-code and ensure that your code is shared in a way that follows best practice and facilitates reproducibility and reuse. 4. We note that your Data Availability Statement is currently as follows: All relevant data are within the manuscript and its Supporting Information files. Please confirm at this time whether or not your submission contains all raw data required to replicate the results of your study. Authors must share the “minimal data set” for their submission. PLOS defines the minimal data set to consist of the data required to replicate all study findings reported in the article, as well as related metadata and methods (https://journals.plos.org/plosone/s/data-availability#loc-minimal-data-set-definition). For example, authors should submit the following data: - The values behind the means, standard deviations and other measures reported;- The values used to build graphs;- The points extracted from images for analysis. Authors do not need to submit their entire data set if only a portion of the data was used in the reported study. If your submission does not contain these data, please either upload them as Supporting Information files or deposit them to a stable, public repository and provide us with the relevant URLs, DOIs, or accession numbers. For a list of recommended repositories, please see https://journals.plos.org/plosone/s/recommended-repositories. If there are ethical or legal restrictions on sharing a de-identified data set, please explain them in detail (e.g., data contain potentially sensitive information, data are owned by a third-party organization, etc.) and who has imposed them (e.g., an ethics committee). Please also provide contact information for a data access committee, ethics committee, or other institutional body to which data requests may be sent. If data are owned by a third party, please indicate how others may request data access. 5. PLOS requires an ORCID iD for the corresponding author in Editorial Manager on papers submitted after December 6th, 2016. Please ensure that you have an ORCID iD and that it is validated in Editorial Manager. To do this, go to ‘Update my Information’ (in the upper left-hand corner of the main menu), and click on the Fetch/Validate link next to the ORCID field. This will take you to the ORCID site and allow you to create a new iD or authenticate a pre-existing iD in Editorial Manager.

**Additional Editor Comments:**

Rewrite abstract and intro to specify task, dataset, model, metrics, and novelty. Add baseline comparisons, explain model architecture, handle class imbalance, discuss Bengali language challenges, interpretability, and justify the contribution clearly.

**Comments from PLOS Editorial Office: **We note that one or more reviewers has recommended that you cite specific previously published works. As always, we recommend that you please review and evaluate the requested works to determine whether they are relevant and should be cited. It is not a requirement to cite these works. We appreciate your attention to this request.

Reviewers' comments:

Reviewer's Responses to Questions

**Comments to the Author**

1. Is the manuscript technically sound, and do the data support the conclusions?

Reviewer #1: Yes

Reviewer #2: Partly

2. Has the statistical analysis been performed appropriately and rigorously? 

Reviewer #1: Yes

Reviewer #2: No

3. Have the authors made all data underlying the findings in their manuscript fully available?

Reviewer #1: Yes

Reviewer #2: No

4. Is the manuscript presented in an intelligible fashion and written in standard English?

Reviewer #1: Yes

Reviewer #2: No

5. Review Comments to the Author

**Reviewer #1: **Abstract needs complete rewrite. Add concrete info on model type, dataset used, accuracy achieved, and contribution.

In intro, Clearly state the objective. Literature is not well synthesized. Review of recent deep learning NLP works for Bengali is missing.

No clear novelty. Stacking models is not new. Authors must explicitly say: “Our novel contribution is…” and justify how this differs from prior Bengali NLP work.

There is no detail on train-test split, feature preprocessing, or hyperparameter settings.

Authors mention base learners but don’t explain architecture or why those were chosen. Provide diagrams or a clear pipeline.

What metric is used for “highest accuracy” (line 91)? Is it F1? Precision?

Add a limitations paragraph.

Replace sentences like “paves the way for further research” (line 108) with real actionable outcomes. Avoid over-generalization.

Needs professional proofreading.

Read these studies to further enrich your study, (https://doi.org/10.1371/journal.pone.0312395) Supports your paper's socioeconomic perspective, especially how economic indicators shape inequality using ML. (https://doi.org/10.1371/journal.pone.0303883) Demonstrates use of ML with economic indicators in the Sri Lankan context, aligning with your economic modeling. (https://doi.org/10.1371/journal.pone.0282847) Reinforces your use of hybrid models like ANFIS and soft computing in Sri Lankan environmental data modeling.

**Reviewer #2: **Navigating Tenses in Bengali Sentences: A Stacked Ensemble Model for Enhanced

Prediction

1. You mention broad goals around improving accuracy and language modeling for

Bengali, but you never define the specific task (e.g., classification? sentiment?

named entity recognition?) in clear terms at the beginning. Define the target NLP

task explicitly and connect it to your broader motivation, whether it's education,

policy, or social impact in Bengali-speaking regions.

2. You claim your stacked model performs better, but there’s no comparison with a

baseline model (e.g., logistic regression, SVM, or even a single deep model).

This makes it impossible to assess whether the stacking architecture adds real

value. Include results from simpler models to show performance gains.

3. There’s no mention of whether the dataset is balanced. If your dataset has

skewed classes (which is common in real-world Bengali text), then accuracy

alone is misleading. Did you use class weights, oversampling, or SMOTE? This

should be discussed and reflected in metrics like F1 or recall per class.

4. Beyond noting the language, you don’t discuss the linguistic characteristics of

Bengali that make this NLP task uniquely challenging (e.g., rich morphology, low-

resource nature, ambiguous syntax). Add some language-specific insights to

justify why your model is suitable and necessary.

5. Given that your stacked model is complex, how do you explain or interpret its

predictions? Even a brief mention of SHAP values, attention weights, or feature

importance would increase trust and make the model more actionable for

policymakers or users in the region.

6. Refer to this work, relevant to your socio-contextual motivation of language-

based emotion/sentiment classification in noisy settings.

DOI: 10.1038/s41598-024-67269-2

6. PLOS authors have the option to publish the peer review history of their article (what does this mean?). If published, this will include your full peer review and any attached files.

Reviewer #1: No

Reviewer #2: No

---

## [Author Response · Author response to Decision Letter 1]

9 Jun 2025

Title: “Navigating Tenses in Bengali Sentences: A Stacked Ensemble Model for Enhanced

Prediction” (Submission ID: PONE-D-24-57203)

# Response to Reviewer 1: (Color code= Red)

#Point 1:

Abstract needs complete rewrite. Add concrete info on model type, dataset used, accuracy

achieved, and contribution.

Author Response:

Thank you for your valuable and constructive feedback. In response to your comment, we have completely

revised the abstract to clearly articulate the scope, methodology, and contributions of our work. The updated

abstract now specifies the NLP task of Bengali tense classification and its relevance to downstream

applications. We describe the construction of a novel dataset, “BengaliTenseCorpus,” comprising 13,500

manually labeled sentences across three tense categories. Additionally, we detail our proposed stacked

ensemble model integrating five base classifiers (Random Forest, SVM, XGBoost, LSTM, and GRU) with

a neural network meta-learner, and report the achieved test accuracy of 85%. We also highlight the novelty

of combining machine learning and deep learning techniques for this task and outline future directions.

These additions ensure the abstract conveys the model type, dataset used, accuracy obtained, and the overall

contribution of our work, in line with your suggestion.

Author Action:

Based on the reviewer’s comments, we have updated our Abstract Section to meet the reviewer’s

concerns.

#Point 2:

In intro, Clearly state the objective. Literature is not well synthesized. Review of recent deep

learning NLP works for Bengali is missing.

Author Response:

Thank you for your thoughtful feedback. In response, we have revised the introduction to explicitly state

the main objective of the study as developing a robust stacked ensemble model for automatic tense

classification in Bengali sentences and its significance in the context of Bangla NLP. Additionally, we

have strengthened the literature review by synthesizing existing works more cohesively and

incorporating several recent studies that apply deep learning techniques to Bengali language tasks, such

as classification, prediction, and detection. These additions provide a clearer context for our contribution

and ensure that the background is both comprehensive and current. We believe these revisions directly

address the reviewer’s concerns and enhance the clarity and relevance of the introduction.

Author Action:

Based on the reviewer’s comments, we have updated our Introduction Section and Literature

Review Section to meet the reviewer’s concerns.#Point 3:

No clear novelty. Stacking models is not new. Authors must explicitly say: “Our novel contribution

is…” and justify how this differs from prior Bengali NLP work.

Author Response:

Thank you for your kind remarks. There is little to no work present in the predictive model to

predict the tense of Bangla text. The sole reason for this problem is the lack of a valid dataset. Our

novel contribution is to create a meaningful dataset for Bangla tense prediction and create a

predictive model around it

Author Action:

We have included our novel contribution statement at the end of the introduction section according to

the reviewer's suggestions.

#Point 4:

There is no detail on train test split, feature preprocessing, or hyperparameter settings.

Author Response:

Thank you for your valuable feedback.

Author Action:

With the renamed subsection of "Model Selection and Hyper Parameters," all information

pertaining to the train-test split and hyper hyperparameter setup is added to the methodology

section. Likewise, details regarding feature processing are included in the methodology part of the

paragraph titled "Data Pre-processing".

#Point 5:

Authors mention base learners but don’t explain architecture or why those were chosen. Provide

diagrams or a clear pipeline.

Author Response:

We selected base models based on their empirical performance during cross-validation. Each

model demonstrated strong predictive ability individually, which makes them suitable candidates

for contributing diverse and accurate predictions in the ensemble. Our ensemble model consists of

machine learning models and deep learning models. These models use different learning biases.

This diversity is essential in stacking to ensure the meta-learner can generalize well across

prediction errors. A clear pipeline is presented in Figure 10.

Author Action:

A brief discussion of our preferred models while creating the ensemble model is given in “Our Model”

subsection.

#Point 6:

What metric is used for “highest accuracy” (line 91)? Is it F1? Precision?

Author Response:

Thank you for your feedback. The accuracy score of such a model is referred to as the highest

accuracy. For that model, the F1 score, precision, and recall were all 0.989.Author Action:

We have rewritten the sentence accordingly.

#Point 7:

Add a Limitation Paragraph.

Author Response:

We appreciate reviewers' valuable insight.

Author Action:

A paragraph has been added at the end of the conclusion section to discuss the limitations and future

work.

#Point 8:

Replace sentences like “paves the way for further research” (line 108) with real actionable

outcomes. Avoid over generalization. Needs professional proofreading.

Author Response:

In response, we have revised overgeneralized statements such as “paves the way for further

research” (line 108) and replaced them with specific, actionable outcomes related to dataset

expansion, transformer-based model integration, and tense morphology conversion. Additionally,

the entire manuscript has been carefully proofread and refined to improve clarity, precision, and

professionalism in language. We believe these changes enhance both the quality and specificity of

the manuscript.

Author Action:

Based on the reviewer’s comments, we have replaced the statement with constructive outcome.

#Point 9:

Read these studies to further enrich your study, (https://doi.org/10.1371/journal.pone.0312395)

Supports your paper's socioeconomic perspective, especially how economic indicators shape

inequality using ML. (https://doi.org/10.1371/ journal.pone.0303883) Demonstrates use of ML

with economic indicators in the Sri Lankan context, aligning with your economic modeling.

(https://doi.org/10.1371/journal.pone.0282847) Reinforces your use of hybrid models like ANFIS

and soft computing in Sri Lankan environmental data modeling.

Author Response:

Thank you for recommending these valuable studies. We have reviewed all three articles and

incorporated relevant insights into the manuscript to strengthen the socioeconomic and

methodological context of our work. Specifically, we referenced the use of machine learning in

economic inequality modeling and hybrid approaches like ANFIS in environmental data analysis,

particularly in low-resource contexts similar to ours. These additions have enriched the

background and discussion sections and helped to better position our contribution within existing

literature.

Author Action:

Based on the reviewer’s comments, we have updated the literature review.# Response to Reviewer 2: (Color code= Green)

#Point 1:

You mention broad goals around improving accuracy and language modeling for Bengali, but you

never define the specific task (e.g., classification? sentiment? named entity recognition?) in clear

terms at the beginning. Define the target NLP task explicitly and connect it to your broader

motivation, whether it's education, policy, or social impact in Bengali speaking regions.

Author Response:

In response, we have revised the introduction to clearly define the target NLP task as automatic tense

classification in Bengali text. We have explicitly stated this task early in the introduction and connected it

to broader motivations such as improving educational tools, supporting grammar correction, and enabling

more inclusive and effective NLP technologies for Bengali-speaking communities. This clarification helps

to better frame the scope and societal relevance of our work.

Author Action:

Updated the introduction to explicitly define the target NLP task as automatic tense classification

and linked it to broader motivations such as educational support, grammar correction, and

enhancing NLP applications for Bengali-speaking communities.

#Point 2:

You claim your stacked model performs better, but there’s no comparison with a baseline model

(e.g., logistic regression, SVM, or even a single deep model). This makes it impossible to assess

whether the stacking architecture adds real value. Include results from simpler models to show

performance gains.

Author Response:

We thank the reviewers for their valuable insights. We included a detailed comparison table of

several machine learning models (6 models) and deep learning (3 models) models' performance in

the “Results and Discussion” section, with accuracy, recall, and F-1 scores. In every case, our

proposed stacking ensemble model outperformed all the machine learning and deep learning

models.

Author Action:

We included a paragraph stating the information in the “Proposed Combined Model” section.#Point 3:

There’s no mention of whether the dataset is balanced. If your dataset has skewed classes (which

is common in real world Bengali text), then accuracy alone is misleading. Did you use class

weights, oversampling, or SMOTE? This should be discussed and reflected in metrics like F1 or

recall per class.

Author Response:

Thank you for this valuable observation. Our dataset was balanced. For that reason, we didn’t

require any oversampling techniques. The support value that reflects the per-class instance is

present in every table for every model in the “Results and Discussion” section.

Author Response:

We have also included the dataset information about balancing in “Dataset Collection and

Properties” subsection.

#Point 4:

Beyond noting the language, you don’t discuss the linguistic characteristics of Bengali that make

this NLP task uniquely challenging (e.g., rich morphology, low resource nature, ambiguous

syntax). Add some language specific insights to justify why your model is suitable and necessary.

Author Response:

Thank you for this valuable observation. In response, we have revised the introduction to include

a detailed discussion of Bengali’s linguistic characteristics that make tense classification

particularly challenging. Specifically, we address its rich inflectional morphology, low-resource

nature, syntactic ambiguity, and the predominance of suffix-based tense marking. We also explain

how our stacked ensemble model is well-suited to handle these complexities by combining deep

learning for sequence modeling and machine learning for robust pattern recognition. These

additions help clarify the necessity and relevance of our approach

Author Action:

We have Revised the introduction to include a detailed explanation of Bengali's linguistic features,

such as its rich morphology, low-resource status, and syntactic ambiguity, and justified the

suitability of the proposed stacked ensemble model in addressing these challenges.

#Point 5:

Given that your stacked model is complex, how do you explain or interpret its predictions? Even

a brief mention of SHAP values, attention weights, or feature importance would increase trust and

make the model more actionable for policymakers or users in the region.

Author Response:

We appreciate the reviewer’s emphasis on model interpretability, particularly given the complexity

of our stacked ensemble and the importance of actionable insights for policymakers. While SHAP

analysis is challenging to apply directly to our current ensemble due to the heterogeneous natureof the base models and their text processing pipelines, we agree that interpretability is critical for

transparency and trust. In future work, we aim to incorporate token-level attribution methods such

as SHAP.

Author Action:

Added as Future Work in the Conclusion section.

#Point 6:

Refer to this work, relevant to your socio contextual motivation of language based

emotion/sentiment classification in noisy settings. DOI: 10.1038/s41598-024-67269-2

Author Response:

In response, we have tried to reach this paper. We found a similar paper titled “Customer Sentiment

Recognition in Conversation Based on Contextual Semantic and Affective Interaction

Information”

Author Action:

We have included the paper in Literature Review section

Again, thank you for your kind cooperation and consideration.

Best Regards,

Md. Nahid Hasan

Corresponding Author

---

## [Decision Letter · Decision Letter 1]

29 Jul 2025

Navigating Tenses in Bengali Sentences: A Stacked Ensemble Model for Enhanced Prediction

PONE-D-24-57203R1

Dear Dr. Hasan,

We’re pleased to inform you that your manuscript has been judged scientifically suitable for publication and will be formally accepted for publication once it meets all outstanding technical requirements.

Kind regards,

Saman Kasmaiee, Ph.D.

Academic Editor

PLOS ONE

Additional Editor Comments (optional):

Reviewers' comments:

Reviewer's Responses to Questions

**Comments to the Author**

1. If the authors have adequately addressed your comments raised in a previous round of review and you feel that this manuscript is now acceptable for publication, you may indicate that here to bypass the “Comments to the Author” section, enter your conflict of interest statement in the “Confidential to Editor” section, and submit your "Accept" recommendation.

Reviewer #3: All comments have been addressed

2. Is the manuscript technically sound, and do the data support the conclusions?

Reviewer #3: Yes

3. Has the statistical analysis been performed appropriately and rigorously? 

Reviewer #3: Yes

4. Have the authors made all data underlying the findings in their manuscript fully available?

Reviewer #3: Yes

5. Is the manuscript presented in an intelligible fashion and written in standard English?

Reviewer #3: Yes

6. Review Comments to the Author

Reviewer #3: The authors have done a good job in addressing the raised issues. It would be great if they could address the issue of inter-annotator agreement score for the manual labelling and error analysis, and also compress teh "preprocessing" section.

7. PLOS authors have the option to publish the peer review history of their article (what does this mean?). If published, this will include your full peer review and any attached files.

Reviewer #3: No

---

## [Editor Report · Acceptance letter]

PONE-D-24-57203R1

PLOS ONE

Dear Dr. Hasan,

I'm pleased to inform you that your manuscript has been deemed suitable for publication in PLOS ONE. Congratulations! Your manuscript is now being handed over to our production team.

Kind regards,

on behalf of

Dr. Saman Kasmaiee

Academic Editor

PLOS ONE